# Shrek MCMC: A Multi-Fidelity Layered MCMC Approach

## Abstract

Markov chain Monte Carlo (MCMC) requires only the ability to evaluate the likelihood, making it a common technique for inference in complex models. However, it can have a slow mixing rate, requiring the generation of many samples to obtain good estimates and an overall high computational cost. Shrek MCMC is a multi-fidelity layered MCMC method that exploits lower-fidelity approximations of the true likelihood calculation to improve mixing and leads to overall faster performance. Such lower-fidelity likelihoods are commonly available in scientific and engineering applications where the model involves a simulation whose resolution or accuracy can be tuned. Our technique uses recursive, layered chains with simple layer tuning; it does not require the likelihood to take any form or have any particular internal mathematical structure. We demonstrate experimentally that Shrek MCMC achieves larger effective sample sizes for the same computational time across different scientific domains including hydrology and cosmology.

## 1 Introduction

Markov chain Monte Carlo (MCMC) is a workhorse of scientific and engineering computation. Most frequently, it is employed to compute the posterior distribution of model parameters, based on observations. The calculated distributions (as represented by samples) give estimates that can be used in calibration and uncertainty quantification to aid in the generation of new scientific experiments, clarify the observability of the model, and resolve scientific theories.

Among the many MCMC algorithms, Metropolis-Hastings MCMC (MH-MCMC) is popular because of its ability to sample from almost any distribution while requiring only the ability to evaluate the model's likelihood given a parameter setting. Yet, this is also its weakness, as it has no additional knowledge of the problem setting to guide its sampling effectively. Therefore, its mixing time (speed of generating effectively new samples) can be slow and the overall algorithm computationally burdensome.

Methods such as Hamiltonian Monte Carlo and its variants (Duane et al., 1987; Neal, 1996; Hoffman & Gelman, 2014) speed up mixing by adding auxiliary momentum variables, allowing longer steps to reduce correlations between consecutive samples. Such methods require computing the gradient of the log target distribution with respect to the parameters, something that could be prohibitively expensive when the distribution is evaluated through lengthy simulation code. For instance, the cosmological simulation we use in our experimental results that aims to approximate the posterior density conditioned on the galaxy power spectrum from DSS-III Baryon Oscillation Spectroscopic Survey (BOSS) Data (Dawson et al., 2013; Alam et al., 2017) cannot be modified to produce gradients. Due to the complexity and non-differentiability of the forward cosmological simulation, gradients with respect to the model parameters are not available. Therefore, methods like auto-differentiation cannot be applied, nor is there an analytic form for the gradients, prohibiting the use of gradient-based inference methods.

Shrek MCMC speeds up the mixing time of MH-MCMC by exploiting lower-fidelity models of the same problem. Many engineering or scientific computational models can be run at multiple fidelities (by tuning a temporal or spatial resolution, for example). By recursively employing MCMC chains, we can use the coarser resolution models to guide the higher resolution MCMC chain. The result is a sampler for the target model that converges faster and generates more effective samples per computation time, even considering the extra time necessary to employ the lower-fidelity computations.

## 2 Background

Markov chain Monte Carlo is a class of algorithms designed to sample from a complicated target distribution by constructing an easy-to-simulate Markov chain such that the stationary distribution of the Markov chain is the target distribution. Commonly, this target distribution is the posterior distribution of a set of parameters, conditioned on observations. Let $D$ be the observations and $\theta \in \Theta \subset \mathbb{R}^R$ be the parameters. Assuming a prior distribution on the parameters $p(\theta)$, the target posterior distribution of interest, $\pi(\theta \mid D)$, is obtained through Bayes' theorem:

$$\pi(\theta \mid D) = \frac{\mathcal{L}(D \mid \theta)p(\theta)}{p(D)} \propto \mathcal{L}(D \mid \theta)p(\theta) \tag{1}$$

where $\mathcal{L}(D \mid \theta)$ is the likelihood of the data, which in many scientific applications requires a lengthy simulation to evaluate. We only require the ability to evaluate $\pi(\theta \mid D)$ up to a constant of proportionality, and therefore the denominator of $p(D)$ is safely ignored. That $\pi(\theta \mid D)$ is a conditional distribution is largely irrelevant for MCMC, so we will just let $\pi(\theta)$ denote the distribution of interest (equal to $\pi(\theta \mid D)$ if the underlying distribution is a posterior, but it could be any distribution over $\theta$).

### 2.1 Metropolis-Hastings MCMC

We focus on the Metropolis-Hastings method for MCMC (MH-MCMC) (Hastings, 1970). The $(i{+}1)$th sample, $\theta^{i+1}$, is generated based on the previous sample in the chain, $\theta^i$, in a two-step process. First, a proposed next state, $\tilde{\theta}^i$ is generated from a proposal distribution, $q(\tilde{\theta}^i|\theta^i)$. Then, $\tilde{\theta}^i$ is either accepted or rejected as $\theta^{i+1}$ according to a carefully constructed acceptance probability. If accepted, $\theta^{i+1}=\tilde{\theta}^i$, otherwise $\theta^{i+1}=\theta^i$. Often, a normal distribution centered at $\theta^i$ is used as the proposal distribution $q(\tilde{\theta}^i|\theta^i)$, but almost any proposal distribution can be used, subject to mild conditions (for instance, that $q(\tilde{\theta}^i|\theta^i)$ is positive everywhere). With a chosen $q(\tilde{\theta}^i|\theta^i)$, the acceptance probability, $\mathcal{A}$, for the transition $\theta^i \to \tilde{\theta}^i$ is

$$\mathcal{A}(\theta^i \to \tilde{\theta}^i) = \min(1, r(\theta^i \to \tilde{\theta}^i)) \tag{2}$$

where

$$r(\theta^i \to \tilde{\theta}^i) = \frac{\pi(\tilde{\theta}^i)}{\pi(\theta^i)} \frac{q(\theta^i|\tilde{\theta}^i)}{q(\tilde{\theta}^i|\theta^i)} \ . \tag{3}$$

Although the standard Metropolis-Hastings MCMC algorithm can be an easy way to sample from a posterior distribution, it requires sufficient samples to be an effective approximation of the posterior distribution. When the chain is slow to mix (due to a less-than-optimal proposal distribution), consecutive samples are highly dependent and more samples must be taken to achieve a set representative of the true distribution. When the evaluation of $\pi(\theta^i)$ (necessary for the calculation of Equation 3) is computationally expensive, this is particularly problematic.

### 2.2 Related Work

Like Shrek MCMC, methods such as simulated tempering and coupled MCMC (Swendsen & Wang, 1986; Marinari & Parisi, 1992; Altekar et al., 2004) use multiple chains. Samples are accepted or rejected by evaluating the energy of the process and adjusting the temperature of the model. Two chains are run in parallel at different temperatures, and the system swaps between different temperatures. Reversible jump MCMC (Green, 1995; Al-Awadhi et al., 2004) also jumps between chains (of different dimensions). While Shrek MCMC shares the notion of multiple chains, because it solves a different problem (to take advantage of simulations that are orders of magnitude cheaper to evaluate), the resulting structure is very different. Methods such as sequential MCMC or particle filtering (Liu & Chen, 1998; Doucet et al., 2001) use the notion of approximations of the target by a large number of samples called particles that are propagated across time using importance sampling. However, those are filtering frameworks and do not converge to a stationary distribution. Thus, though appearing related in its structure, Shrek MCMC is quite different to these methods.

More similar to Shrek MCMC, several previous methods have shown that replacing the proposal with an approximation with generally high acceptance probability reduces the computational cost of the standard

Metropolis-Hastings algorithm significantly. This idea was first proposed by Christen & Fox (2005); Fox & Nicholls (1997) as a two-stage MCMC method that tests the original proposal using a cheap approximation to find moves in the chain that are more likely to be accepted. In other words, a candidate is accepted with the likelihood of the approximate model before it is evaluated with the more expensive model. In preconditioned MCMC using coarse-scale simulation proposed by Efendiev et al. (2006), two-stages are used to reduce the computational cost incurred in the fine fidelity by testing the coarse model based on high-fidelity multiscale finite volume model. However, this only performs a single check with a cheap approximation, and does not exploit it to run a full MCMC subchain.

Multilevel Markov chain Monte Carlo (MLMCMC) (Dodwell et al., 2015) extended this idea taking inspiration from the multilevel Monte Carlo method (Heinrich, 2001) for high-dimensional, parameter-dependent integrals and Multilevel Monte Carlo Path Simulation (Giles, 2008) originally in the context of stochastic differential equations in finance by subsampling at coarser levels: If the coarse proposal from the approximation is rejected by the fine level, the coarse chain continues independently of the fine chain instead of recursively starting the next coarse chain from the current sample of the fine chain. MLMCMC uses a user-specified variable that is internal to the likelihood computation and shared across the levels (for instance, the predicted observations to be compared with the true observations through a noise model). The samples drawn from the coarse approximation are used to reduce the variance of this internal variable achieving better proposals from the coarse fidelities.

Lykkegaard et al. (2023) proposed Adaptive Multilevel Delayed Acceptance (MLDA), which adapted a recursive version of MLMCMC over multiple levels. Here, the coarse inner subchain used to generate subsequent proposals for the current chain is initiated from the current sample from the outer chain again instead of independently continuing the fine chain even if coarse proposal is rejected. MLDA also applies an Adaptive Error Model (AEM) (Kaipio & Somersalo, 2007) to account for discrepancies between the different fidelities. It takes the two-level AEM from Adaptive Delayed Acceptance Metropolis Hastings (Cui et al., 2012; 2019) and extends it by adding a telescoping sum of differences in the model output across multiple levels.

Several multilevel MCMC methods based on delayed rejection, in contrast to delayed acceptance, have also been proposed and are summarized by Peherstorfer et al. (2018). Adaptive methods in multistage MCMC Tierney & Mira (1999) proposed using an independence sampler that is a good approximation for the posterior distribution in the first stage and random walk in the second stage to help with poor approximation by the independence sampler. Delayed rejection in MCMC (Green & Mira, 2001) suggested using a normal distribution as the proposal in the first level and a normal distribution with the same mean but higher variance in the second level. Higdon et al. (2002) proposed using multiple MCMC chains from low and high fidelities and coupling them using a product chain and "swapping" updates allowing information to move between the two fidelity scales. Conrad et al. (2016) uses local approximations of either the log-likelihood function or the forward model of different simulations into the Metropolis-Hastings kernel. Although these methods use approximations as proposals, they do not exploit layered or recursive MCMC chains.

Cai & Adams (2022) proposed a multi-fidelity Monte Carlo method (MFMC) that uses randomized fidelities as the approximate for the target fidelity. The algorithm does not converge to the true posterior, but the resulting samples can be used to estimate expectations through a specific "sign-correction" formula. Our method follows a hierarchy of levels in its sampling while also sampling from the true posterior and provides a simpler alternative to previous multilevel methods.

## 2.3 Our Contributions

Our multi-fidelity layered MCMC algorithm, Shrek MCMC, has a similar structure to MLDA in terms of the recursive layers and achieves a similar amount of effective samples across multiple chains of MLDA. However, our method for mitigating the differences between approximations is simpler in construction and implementation than that of MLDA, does not require the identification of any internal variables of the distribution to be sampled, and generates more effective samples in a shorter amount of time and computational cost. We demonstrate this on real-world large scientific problems. We also show theoretical convergence rates and prove ergodicity of the adaptation in layer tuning.

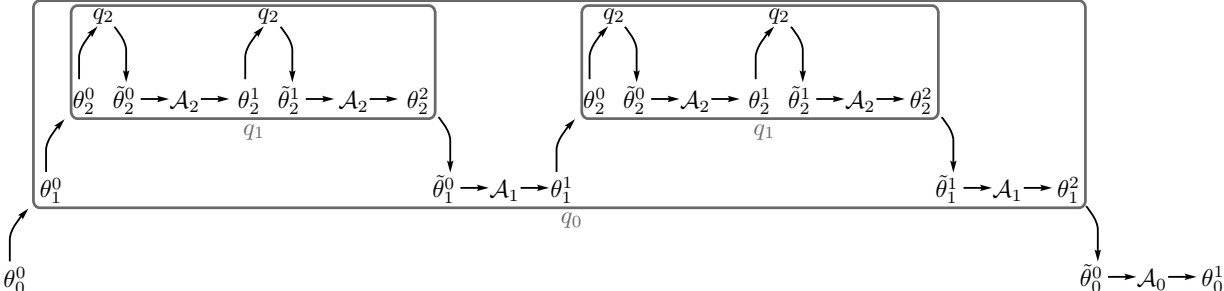

Figure 1: One sampling step from the finest layer with two coarse fidelities and two iterations per nested chain. Refer to text in Section 3.1.

# 3   Multi-Fidelity Layered MCMC: Shrek MCMC

We consider a series of models, ordered by fidelity. For instance, we might have a model that evaluates a differential equation numerically as the main part of the likelihood calculation (simulating forward in time); the resolution of the spatial or temporal grid used to evaluate the model can be tuned to change its fidelity. The highest fidelity model is our "true" model, from whose posterior we wish to sample. Shrek MCMC draws samples from the true model. Its nested chains use the coarser fidelity models as cheap approximations of this finest fidelity model to speed up mixing.

Where the standard Metropolis-Hastings algorithm uses a distribution $q$ that proposes the next sample, Shrek MCMC uses nested Markov chains as the proposal distribution. In a recursive fashion, each layer uses the result of another MCMC chain with a coarser approximation as its proposal. The recent sample in the current chain is the starting sample in the nested chain. The coarser chain runs for $M$ iterations, with each proposed sample evaluated by the likelihood of the cheaper layer. The last, $M$th, sample of the coarser chain is proposed as the candidate for the next sample in the current chain. At the coarsest fidelity/layer, a standard proposal distribution is used, for instance a normal distribution centered on the current point.

This avoids numerous expensive likelihood calculations in the fine fidelity that might end up rejected, and it allows the proposal to generate samples that are more likely to get accepted by the finest fidelity, since it was accepted by an approximation already. While the coarser chains have their own computational cost, they can often be orders of magnitude faster to evaluate, thus leading to an overall savings in the running time of the entire algorithm, as measured by the quality of the samples generated per computational time.

## 3.1   Algorithm Specification

Let $\theta \in \mathbb{R}^R$ be the set of parameters (over which we are sampling) and let $j \in \{0, 1, ..., J\}$ be the fidelities ordered in a decreasingly complex fashion (0 is the "true" model and $J$ is the coarsest fidelity). We let $\pi_j(\theta)$ be the posterior distribution according to the $j$th fidelity model and $q_j(\tilde{\theta}|\theta)$ be the proposal distribution for layer $j$. Here, $\theta_j^i$ is the $i$th sample in the current chain at layer $j$. The goal is to sample from $\pi_0(\theta)$.

In Shrek MCMC, the proposal distribution $q_j(\tilde{\theta}_j^i|\theta_j^i)$ for iteration $i$ of a chain at layer $j$ is another MCMC chain of $M$ steps targeting the (coarser) posterior $\pi_{j+1}(\cdot)$, starting this nested chain at $\theta_j^i$. The result of $M$ steps using a chain with stationary distribution $\pi_{j+1}(\cdot)$ is the proposal for $\tilde{\theta}_j^i$: $q_j(\tilde{\theta}_j^i|\theta_j^i)$. More algorithmically, to generate $\tilde{\theta}_j^i$ from $\theta_j^i$, we run the (coarser) MCMC algorithm at layer $j+1$. We start with $\theta_{j+1}^0 = \theta_j^i$ and continue the coarser MCMC sampler until $\theta_{j+1}^M$. We then set $\tilde{\theta}_j^i = \theta_{j+1}^M$. At the coarsest layer, $q_J(\tilde{\theta}^i|\theta^i)$ is a standard simple proposal distribution. Figure 1 pictorially demonstrates this for $J = 2$ inner layers, each with $M = 2$ steps.

With the sampling scheme so defined, it remains to construct the acceptance probability for each layer: $\mathcal{A}_0, \mathcal{A}_1, \ldots, \mathcal{A}_J$. We follow a standard Metropolis-Hastings method for every layer and therefore $\mathcal{A}_j = \min(1, r_j(\theta_j^i \rightarrow \tilde{\theta}_j^i))$. At the coarsest layer, the ratio $r_J(\theta_J^i \rightarrow \tilde{\theta}_J^i)$ is just as in Equation 3 because $q_J$ is a standard proposal distribution.

---

**Algorithm 1** ShrekChain($\theta_j^0$,n,$j$)

---

1: **for** $i = 0, \ldots, n - 1$ **do**
2:     **if** $j = J$ **then**                                                     ▷ coarsest layer
3:         Sample $\tilde{\theta}_j^i$ from $q_j(\cdot|\theta_j^i)$
4:         Accept $\theta_j^{i+1} = \tilde{\theta}_j^i$ with probability $\mathcal{A}$ from Equation 2
5:         Otherwise, reject and $\theta_j^{i+1} = \theta_j^i$
6:     **else**
7:         $\theta_{j+1}^1, \ldots, \theta_{j+1}^M = $ ShrekChain($\theta_j^i$, M, $j{+}1$)
8:         $\tilde{\theta}_j^i = \theta_{j+1}^M$
9:         Accept $\theta_j^{i+1} = \tilde{\theta}_j^i$ with probability $\mathcal{A}_j$ from Equation 5
10:        Otherwise, reject and $\theta_j^{i+1} = \theta_j^i$
11: **return** $\theta_j^1, \ldots, \theta_j^n$

---

When $j < J$, the proposal distribution is from a Markov chain that obeys detailed balance. Therefore

$$\frac{q_{j+1}(\theta_j^i|\tilde{\theta}_j^i)}{q_{j+1}(\tilde{\theta}_j^i|\theta_j^i)} = \frac{\pi_{j+1}(\theta_j^i)}{\pi_{j+1}(\tilde{\theta}_j^i)} \qquad 0 \le j < J \tag{4}$$

and thus

$$\mathcal{A}_j(\theta_j^i \to \tilde{\theta}_j^i) = \min\left(1, \frac{\pi_j(\tilde{\theta}_j^i)}{\pi_j(\theta_j^i)} \cdot \frac{\pi_{j+1}(\theta_j^i)}{\pi_{j+1}(\tilde{\theta}_j^i)}\right) . \tag{5}$$

Note this equation does not depend on $M$ (the number of steps for the coarser chain at layer $j + 1$). While this chain has almost certainly not mixed for small $M$, the ratio $q_j(\theta_j^i|\tilde{\theta}_j^i)/q_j(\tilde{\theta}_j^i|\theta_j^i)$ is the same as if the chain had completely mixed and the proposed new state, $\tilde{\theta}_j^i$, were from the true posterior of the model at layer $j + 1$. The values $\pi_{j+1}(\tilde{\theta}_j^i)$ and $\pi_{j+1}(\theta_j^i)$ were already calculated as part of the chain at layer $j + 1$ and therefore do not take any additional computation time. Shrek MCMC is summarized in Algorithm 1. To gather $N$ samples from the true posterior, the algorithm is called as $ShrekChain(\theta^0, N, J = 0)$.

### 3.2 Convergence

Shrek MCMC convergence is assured through standard Metropolis-Hastings convergence:

**Definition 3.1.** *Proposal distribution $q_j(\cdot|\cdot)$ has **full support** iff $(\pi_j(\tilde{\theta}) > 0$ and $\pi_j(\theta) > 0) \to (q_j(\tilde{\theta}|\theta) > 0)$.*

**Lemma 3.1.** *If the support of $\pi_{j+1}(\cdot)$ is a superset of the support of $\pi_j(\cdot)$ (i.e., $\pi_j(\theta) > 0$ only if $\pi_{j+1}(\theta) > 0$) and $q_{j+1}(\cdot|\cdot)$ has full support, then $q_j(\cdot|\cdot)$ has full support.*

*Proof.* Because $q_{j+1}(\cdot|\cdot)$ has full support, the support of $q_{j+1}(\cdot|\theta_j^i)$ is a superset of $\pi_{j+1}(\cdot)$ for all $\theta_j^i$ that would be considered. This further implies that there is a non-zero chance of accepting the proposed state for all proposals within the support of $\pi_{j+1}(\cdot)$. Because the support of $\pi_{j+1}(\cdot)$ is a superset of the support of $\pi_j(\cdot)$, this means that $q_j(\cdot|\cdot)$ similarly has full support. □

**Theorem 3.1.** *If $q_J(\cdot|\cdot)$ has full support, and $\pi_j(\cdot)$ has the same support for all $0 \le j \le J$, then Shrek MCMC converges to samples from $\pi_0(\cdot)$.*

*Proof.* If $q_J(\cdot|\cdot)$ has full support, by Lemma 3.1 and induction, $q_0(\cdot|\cdot)$ has full support. The acceptance ratio for level 0 is constructed exactly according to the Metropolis-Hastings formula and therefore obeys detailed balance. Because the proposal distribution has full support, the chain is ergotic. Together with standard Metropolis-Hastings arguments, this means that Shrek MCMC converges to the target distribution. □

### 3.3 Convergence Rate

We show a convergence rate for Shrek MCMC. We measure the distance to the stationary in terms of total variation distance as follows.

**Definition 3.2.** *The total variation distance between two probability measures $\nu_1(\cdot)$ and $\nu_2(\cdot)$ is*

$$\|\nu_1(\cdot) - \nu_2(\cdot)\| = \sup_A |\nu_1(A) - \nu_2(A)|.$$

The minorization condition of Markov chains, first introduced by Roberts & Rosenthal (2004), can be used to measure convergence rate. For any Markov chain with a one step transition probability of $p(\theta^i \to \theta^{i+1})$, we let $p^n(\theta^i \to \theta^{i+n})$ denote the corresponding $n$ step transition probability. Formally, the definition of the minorization condition is stated below.

**Definition 3.3.** *A Markov chain on $\Theta$ satisfies the minorization condition if there exists an $\epsilon > 0$, a positive integer $n$, and a probability measure $\nu(.)$ such that*

$$p^n(\theta^0 \to \theta^n) \geq \epsilon \nu(\theta^n) \qquad \forall \theta^0, \theta^n \in \Theta. \tag{6}$$

With respect to the stationary distribution, the probability of transitioning from $\theta$ to $\theta'$ can be minorized by a lower bound such that $p_j^1(\theta \to \theta') \geq \epsilon \pi_j(\theta')$.

**Lemma 3.2.** *Assume the minorization condition holds at the innermost level ($j = J$): $p_J^1(\theta_J^i \to \theta_J^{i+1}) \geq \xi_J \cdot \pi_J(\theta_J^{i+1})$ for some $\xi_J > 0$. Then, there exists a minorized lower bound on levels $j < J$ such that*

$$p_j^1(\theta_j^i \to \theta_j^{i+1}) \geq \xi_j \cdot \pi_j(\theta_j^{i+1}) \qquad \forall \theta_j^i, \theta_j^{i+1} \tag{7}$$

*where $\xi_j = (1 - (1 - \xi_{j+1})^M) \cdot \min_\theta \left( \frac{\pi_{j+1}(\theta)}{\pi_j(\theta)} \right)$.*

We call $\xi_j$ the minorization constant for level $j$. This satisfies the necessary minorization condition of Roberts & Rosenthal (2004). We apply Theorem 8 of the same paper. This allows us to get a quantitative bound on the distance to the stationary of every level as stated in Theorem 3.2. The proof for Lemma 3.2 can be found in the Appendix.

**Theorem 3.2.** *Let $p_j^n(\theta_j^0 \to \cdot)$ be the distribution for layer $j$ with an invariant target probability $\pi_j(\cdot)$. Shrek MCMC is uniformly ergodic and converges as $\left\|p_j^n(\theta_j^0 \to \cdot) - \pi_j(\cdot)\right\| \leq (1 - \xi_j)^n$ where $\xi_j = (1 - (1 - \xi_{j+1})^M) \cdot \min_\theta \left( \frac{\pi_{j+1}(\theta)}{\pi_j(\theta)} \right)$ as in Lemma 3.2. Most critically, it holds for layer $j = 0$.*

*Proof.* The results follow from the minorization condition established in Lemma 3.2. □

### 3.4 Layer Tuning

The algorithm above uses the coarser fidelities to guide the finer ones. Early in the chain, this is useful for quickly driving the samples toward high-probability regions. However, this mismatch between the fidelities can cause problems later because it can steer the chain away from high-probability regions in the fine fidelity model that do not overlap with high-probability regions of the coarse fidelity model. Figure 2 demonstrates an example of such partial, but not com-

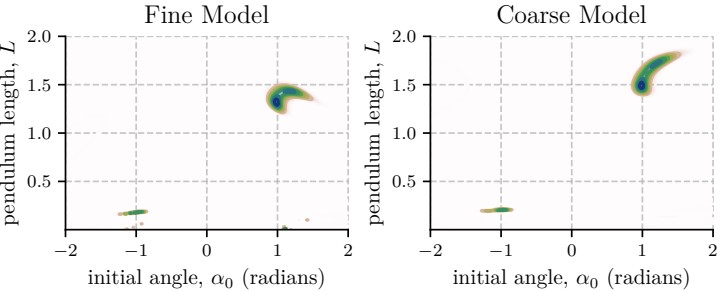

Figure 2: Posterior of different fidelities; coarse model is the small angle approximation. See Section 4.1.

plete, overlap in one of our examples. To combat this, we present a simple modification that does not require estimation of any internal variables of the probability models, nor estimation of means or variances from multiple chains.

Recall $\pi_j(\theta_j)$ is known up to a normalizing constant: $\pi_j(\theta_j) = \tilde{\pi}_j(\theta_j)/Z_j$ where $\tilde{\pi}_j(\theta_j)$ is the unnormalized distribution and $Z_j$ is the normalzing constant. We modify the target distributions for coarser chains (and thus the proposal distributions for all $j > 0$) as

$$\psi_j(\theta_j) = \big(\tilde{\pi}_j(\theta_j) + \omega_j\big)/\zeta_j(\omega_j) \qquad\qquad \forall \ \ 0 < j \leq J \tag{8}$$

where $\zeta_j(\omega_j)$ is the normalizing constant of this new distribution which depends on $\omega_j$.[1]

We now use $\psi_j(\theta_j)$ in place of $\pi_j(\theta_j)$ in Equation 5, therefore modifying the acceptance ratio for all $0 < j \leq J$ as

$$\mathcal{A}_j(\theta_j^i \to \tilde{\theta}_j^i) = \min\left(1, \frac{\psi_j(\tilde{\theta}_j^i)}{\psi_j(\theta_j^i)} \cdot \frac{\psi_{j+1}(\theta_{j+1}^i)}{\psi_{j+1}(\tilde{\theta}_{j+1}^i)}\right) . \tag{9}$$

For the finest layer, things remain the same (or alternatively, $\omega_0 = 0$), because we do not want to change the distribution of the overall sampler.

This effectively mixes the stationary distribution of the $j$th layer with a uniform distribution (we have added a constant to the posterior and then renormalized), encouraging the proposal to explore more widely than the coarser layer would normally. While unsophisticated, we found it simpler to implement and compute than other options and just as effective.

Instead of leaving $(\omega_1, \omega_2, \ldots, \omega_J)$ as hyper-parameters, we use gradient descent to adapt them over the course of the sampling. We adjust $\omega_{j+1}$ to minimize the Kullback-Leibler divergence between layers $\psi_j$ and $\psi_{j+1}$:

$$\mathrm{KL}(\psi_j \| \psi_{j+1}) = \mathop{\mathbb{E}}_{\theta \sim \psi_j} [\ln(\psi_j)] - \mathop{\mathbb{E}}_{\theta \sim \psi_j} [\ln(\psi_{j+1})] . \tag{10}$$

This tries to make the coarser (approximating) distribution $\psi_{j+1}$ more similar to the distribution $\psi_j$. Because the first term does not depend on $\omega_{j+1}$, the objective function is to maximize

$$H_{j+1} = \mathop{\mathbb{E}}_{\theta \sim \psi_j} \left[\ln(\psi_{j+1})\right] . \tag{11}$$

Using Equation 8,

$$\frac{\partial}{\partial \omega_{j+1}} H_{j+1} = \frac{\partial}{\partial \omega_{j+1}} \left( \mathop{\mathbb{E}}_{\theta \sim \psi_j} [\ln\left(\tilde{\pi}_{j+1}(\theta) + \omega_{j+1}\right)] - \ln \zeta_{j+1}(\omega_{j+1}) \right)$$

$$= \mathop{\mathbb{E}}_{\theta \sim \psi_j} \left[ \frac{\partial}{\partial \omega_{j+1}} \ln\left(\tilde{\pi}_{j+1}(\theta) + \omega_{j+1}\right) \right] - \mathop{\mathbb{E}}_{\theta \sim \psi_{j+1}} \left[ \frac{\partial}{\partial \omega_{j+1}} \ln\left(\tilde{\pi}_{j+1}(\theta) + \omega_{j+1}\right) \right] \tag{12}$$

where the second step replaces the derivative of the log-partition function with the expected derivative of the log-probability.

The first term is an expectation with respect to the distribution at the lower layer $j$. We assume that the lower layer has mixed and therefore, the starting state for the chain at layer $j + 1$ is a sample drawn from $\psi_j$. The second term is an expectation with respect to the distribution at this layer, $j + 1$. We let the sample at the *end* of this chain after $M$ steps approximate a sample from this distribution. This is similar to the approximation employed by $M$-step contrastive divergence (Hinton, 2002). Although this is not guaranteed to converge (Sutskever & Tieleman, 2010), in practice we have found it to work well. Thus, the total derivative for the gradient ascent update is

$$\frac{\partial}{\partial \omega_{j+1}} H_{j+1} \approx \frac{1}{\tilde{\pi}_{j+1}(\theta_{j+1}^0) + \omega_{j+1}} - \frac{1}{\tilde{\pi}_{j+1}(\theta_{j+1}^M) + \omega_{j+1}} . \tag{13}$$

Note that these denominators are calculated during the MCMC chain and therefore the derivative requires very little extra computation.

---

**Algorithm 2** ShrekWithLayerTuning($\theta_j^0$,n,$j$)

---

1: **for** $i = 0, \ldots, n-1$ **do**
2:      **if** $j = J$ **then**                                             ▷ coarsest layer
3:          Sample $\tilde{\theta}_j^i$ from $q_j(\cdot|\theta_j^i)$
4:          Accept $\theta_j^{i+1} = \tilde{\theta}_j^i$ with probability $\mathcal{A}$ from Equation 2
5:          Otherwise, reject and $\theta_j^{i+1} = \theta_j^i$
6:      **else**
7:          $\theta_{j+1}^1, \ldots, \theta_{j+1}^M = $ ShrekWithLayerTuning($\theta_j^i$, M, $j$+1)
8:          $\tilde{\theta}_j^i = \theta_{j+1}^M$
9:          Update gradient of $\omega_{j+1}$ using $\frac{\partial}{\partial \omega_{j+1}} H_{j+1}$ from Equation 13
10:         Accept $\theta_j^{i+1} = \tilde{\theta}_j^i$ with probability $\mathcal{A}_j$ from Equation 9
11:         Otherwise, reject and $\theta_j^{i+1} = \theta_j^i$
12: **return** $\theta_j^1, \ldots, \theta_j^n$

---

A single $\omega_j$ is kept for each layer and is maintained across subchains at that layer. We use a learning rate of $10^{-3}$ to adjust $\omega_j$ for all experiments. An update is made on layer $j$ once after each $M$-step subchain.

To make the innermost Gaussian proposal more robust, we adaptively update the covariance of the proposal distribution as initially proposed in the AM algorithm (Haario et al., 2001). We use the history of chains from the coarsest layer $\theta_J^0, \theta_J^1, \ldots, \theta_J^t$ to update the covariance for the inner most proposal distribution. By using all previous states of the coarsest layer, the proposal distribution quickly adapts using the accepted samples. This rapid start of adaptation ensures good mixing in the inner most layer which gives higher quality candidate samples for the finer chains. We show the recursive algorithm with layer tuning adaptation added in Algorithm 2. Here we update the gradient after $M$ steps of each layer and use it in the acceptance probability with $\omega$ mixed in as a uniform distribution to the target distribution.

### 3.5 Ergodicity of Layer Tuning

We show Shrek MCMC with adaptive tuning of the proposals at each layer is ergodic. This can be shown with diminishing adaptation and simultaneous uniform ergodicity.

**Lemma 3.3.** *For layers $0 \leq j < J$, let $\gamma_j \in \Gamma_j$ be the adaptations for the proposal at layer $j$ or the chain at level $j+1$, i.e, $\gamma_j = \omega_{j+1} \leftrightarrow \psi_{j+1}(\theta) \leftrightarrow p_{j+1}(\theta \to \cdot)$ where $\Gamma_j \in \mathbb{R}$ and $\psi_{j+1}$ is the target at layer $j+1$ with the layer tuning adaptation added. Let $p_{j,\gamma_j}(\theta \to \cdot)$ denote the transition distribution of chain at level $j$ using adaptation $\gamma_j$, starting in state $\theta$. Assume $\forall j, \omega_j \in [\underline{\omega}, \overline{\omega}]$ for some $0 < \underline{\omega} < \overline{\omega}$, and, at the inner most layer, there exists a minorization constant $\xi_J > 0$ such that $\left\| p_{J,\gamma_J}^M(\theta \to \cdot) - \psi_J(\cdot) \right\| \leq (1 - \xi_J)^M$. Then,*

*(a) Simultaneous uniform ergodicity: For all $\tau > 0$. there exists $n = n(\tau) \in \mathbb{N}$ such that*

$$\left\| p_{J,\gamma_J}^n(\theta \to \cdot) - \psi_j(\cdot) \right\| \leq \tau \tag{14}$$

*for all $\theta \in \Theta$ and $\gamma_j \in \Gamma_j$.*

*(b) Diminishing adaptation: The amount of adaptation diminishes in probability with the number of steps $t$ in the adaptation as*

$$\lim_{t \to \infty} \sup_\theta \left\| p_{j,\gamma_j^t}(\theta \to \cdot) - p_{j,\gamma_j^{t+1}}(\theta \to \cdot) \right\| = 0. \tag{15}$$

Proof for Lemma 3.3 can be found in the Appendix.

**Theorem 3.3.** *Shrek MCMC with an adaptive layer tuning parameter is ergodic.*

---

[1]We assume the domain of $\theta$, $\Theta$, is of finite volume.

*Proof.* We use Lemma 3.3 to show the conditions necessary in Theorem 1 of Roberts & Rosenthal (2007). This shows that the adaptive algorithm is ergodic. □

## 4 Experiments

We measure the efficiency of the MCMC methods tested using the effective sample size (ESS) (Ripley, 1987) estimated across multiple chains as $N_{ESS} = (N \cdot K) / \left(1 + 2\sum_{k=1}^{2m+1} \rho(k)\right)$ where $N$ is the number of samples, $K$ is the number of chains, $\rho(k)$ is the lag-$k$ correlation, and $m$ is the largest value such that $\rho(2m) + \rho(2m+1) > 0$. We compute ESS for each parameter for the "bulk" (entire distribution) and "tail" (largest and smallest 5% of the samples) of the distributions.

We compare Shrek MCMC with standard Metropolis-Hastings (with proposal adaptation of Haario et al. (2001)) and other multi-fidelity MCMC methods: Multilevel Delayed Acceptance MCMC (MLDA) (Lykkegaard et al., 2023), MLDA with Adaptive Error Model (AEM) (Cui et al., 2012; Lykkegaard et al., 2023), Multi Level MCMC (MLMCMC) (Dodwell et al., 2015; Lykkegaard et al., 2020), and Multi-fidelity Monte Carlo (Cai & Adams, 2022). We give more detail about the different methods used for comparison:

1. MLDA without any adaptation: Introduced by Lykkegaard et al. (2023), this method uses recursive chains of approximations as proposals. However, there is no adaptation being done to "correct" the approximations. Even Lykkegaard et al. (2023) note that without adaptation, the chains do not mix well, and have poor effective sample sizes.

2. MLDA with AEM: Extended by Lykkegaard et al. (2023) in the same paper, this method uses a similar structure to the above. They also use Adaptive Error Model (AEM) as a way to deal with the discrepancies between the different layers which uses a telescoping sum of differences in the mean of the approximations. They demonstrate with the subsurface flow model that MLDA with AEM leads to good mixing and high ESS. We demonstrate similar results for two of our experiments. Our subsurface flow model experiment uses the same fidelities as set up by the original authors; however, we run it to collect more samples using a higher number of chains.

3. Multi Level MCMC (MLMCMC): This method was proposed by Dodwell et al. (2015) and was then applied to MLDA. A quantity of interest, $Q$, is proposed that is related to the parameters of the model. The samples drawn from the posterior are used to reduce the variance of $Q$. Since in MLDA, samples are not only drawn from a "true" posterior, but also approximations, the samples from the approximate levels are used to reduce the variance of $Q$. They state that it thus requires fewer samples to achieve the same variance. Using a telescopic sum, the difference of $Q$ estimates between levels are used to correct $Q$ with respect to the next coarser level. For the pendulum model, $Q$ is the mean of the outputs. For the subsurface flow experiment used by the original authors of MLDA, $Q$ is the hydraulic head at some fixed point $(x, y) = (0.5, 0.45)$; that is, the model PDE is solved at these points at each level using samples from the coarser approximate level.

4. Multi-Fidelity Monte Carlo (MFMC): This method was proposed Cai & Adams (2022). It uses a continuum of models with increasing fidelity and has a single Markov chain with a random choice of the fidelity, $K$, at each step. The fidelity $K$ is part of the sampled state-space and therefore also part of the proposal distribution and acceptance probability. We map $K$ to a reasonable range of fidelities for each experiment. For the pendulum model, we map $K$ to the error tolerance of the integrator, $\epsilon$, as $\epsilon = e^{K/10} + 10^{-6}$. For the subsurface flow experiment, we let the grid resolution be equal to $10K$ ($K$ is the sampled fidelity of this method) in order to map the resolution to the fidelity range expected by the algorithm's implementation.

For the MLDA-based methods, we use the authors' implementations in the open-source probabilistic programming package PyMC3 (Salvatier et al., 2016) by Lykkegaard et al. (2023) For MFMC, we use the author-provided implementation. These implementations have significant computational overhead compared with our implementation of Shrek MCMC. Therefore, we only measure the time taken in likelihood computation (which is the same code for all methods).

Table 1: Mean ESS for bulk and tail distributions across 50 runs for 10 chains each, and mean value of each parameter with std error across all 500 total runs are listed. Average acceptance rates are listed per layer and the total samples are based on the average cost of likelihood evaluations per sample per method. Each chain is run for a total of 1000 seconds.

(a) Simple Pendulum

| | $\alpha$ | | | | $L$ | | | | acceptance rate | | | total samples |
|---|---|---|---|---|---|---|---|---|---|---|---|---|
| | bulk ESS/s | tail ESS/s | mean | sd | bulk ESS/s | tail ESS/s | mean | sd | $j=0$ | $j=1$ | $j=2$ | |
| MCMC | 21.43 | 29.94 | 0.953 | 0.492 | 17.94 | 29.94 | 1.265 | 0.355 | 0.29 | | | 100000 |
| Shrek MCMC(single) | **52.47** | **58.42** | 1.081 | 0.019 | **45.85** | **54.95** | 1.372 | 0.016 | 0.98 | 0.24 | | 60000 |
| AEM MLDA (single) | 39.41 | 53.91 | 1.064 | 0.062 | 42.20 | 52.24 | 1.371 | 0.051 | 0.98 | 0.27 | | 50000 |
| MLMCMC (single) | 25.03 | 34.43 | 1.044 | 0.291 | 21.54 | 32.99 | 1.351 | 0.167 | 0.92 | 0.32 | | 44500 |
| MLDA (single) | 11.64 | 1.49 | 1.059 | 0.041 | 15.36 | 7.73 | 1.361 | 0.022 | 0.91 | 0.31 | | 45000 |
| Shrek MCMC(double) | **64.26** | **72.10** | 1.086 | 0.001 | **56.68** | **68.20** | 1.374 | 0.009 | 0.99 | 0.86 | 0.29 | 35000 |
| AEM MLDA (double) | 56.92 | 65.53 | 1.085 | 0.006 | 50.95 | 58.51 | 1.375 | 0.003 | 0.98 | 0.89 | 0.3 | 25000 |
| MLMCMC (double) | 33.82 | 37.05 | 1.049 | 0.015 | 33.09 | 39.94 | 1.361 | 0.015 | 0.90 | 0.81 | 0.28 | 25500 |
| MLDA (double) | 10.01 | 7.54 | 1.053 | 0.035 | 4.96 | 12.99 | 1.362 | 0.024 | 0.86 | 0.72 | 0.24 | 30000 |
| MFMC | | | 1.075 | 0.057 | | | 1.348 | 0.132 | 0.46 | | | 370000 |

(b) Subsurface Flow model

| | $\theta_1$ | | | | $\theta_2$ | | | | $\theta_3$ | | | | acceptance rates | | | total samples |
|---|---|---|---|---|---|---|---|---|---|---|---|---|---|---|---|---|
| | bulk ESS/s | tail ESS/s | mean | sd | bulk ESS/s | tail ESS/s | mean | sd | bulk ESS/s | tail ESS/s | mean | sd | $j=0$ | $j=1$ | $j=2$ | |
| MCMC | 5.35 | 7.81 | -0.457 | 0.0037 | 5.49 | 8.04 | 0.466 | 0.0036 | 5.63 | 8.19 | 0.076 | 0.0034 | 0.27 | | | 10000 |
| Shrek MCMC (single) | **8.74** | **11.99** | -0.460 | 0.0030 | **8.58** | **11.81** | 0.467 | 0.0036 | **8.63** | **11.58** | 0.076 | 0.0028 | 0.98 | 0.26 | | 8365 |
| AEM MLDA (single) | 7.77 | 4.67 | -0.459 | 0.0032 | 6.96 | 4.29 | 0.466 | 0.0033 | 8.16 | 6.24 | 0.076 | 0.0030 | 0.99 | 0.29 | | 4100 |
| MLMCMC (single) | 4.35 | 5.86 | -0.460 | 0.0028 | 4.37 | 3.27 | 0.465 | 0.0031 | 5.29 | 6.49 | 0.077 | 0.0021 | 0.93 | 0.32 | | 8000 |
| MLDA (single) | 3.65 | 1.52 | -0.463 | 0.0035 | 4.78 | 1.56 | 0.490 | 0.0036 | 3.33 | 3.02 | 0.075 | 0.0031 | 0.87 | 0.34 | | 8250 |
| Shrek MCMC (double) | **15.49** | **21.35** | -0.460 | 0.0026 | **15.141** | **19.69** | 0.469 | 0.0026 | **15.07** | **19.76** | 0.077 | 0.0023 | 0.99 | 0.95 | 0.3 | 6500 |
| AEM MLDA (double) | 13.29 | 14.70 | -0.460 | 0.0035 | 12.32 | 13.22 | 0.468 | 0.0035 | 12.70 | 14.45 | 0.077 | 0.0033 | 0.99 | 0.93 | 0.24 | 2285 |
| MLMCMC (double) | 8.46 | 10.49 | -0.459 | 0.0027 | 8.04 | 9.13 | 0.468 | 0.0030 | 7.47 | 5.81 | 0.077 | 0.0024 | 0.92 | 0.89 | 0.31 | 6000 |
| MLDA (double) | 4.48 | 5.05 | -0.461 | 0.0036 | 5.33 | 6.60 | 0.469 | 0.0033 | 5.28 | 7.35 | 0.076 | 0.0030 | 0.93 | 0.91 | 0.27 | 6350 |
| MFMC | | | -0.475 | 0.026 | | | 0.418 | 0.041 | | | 0.102 | 0.003 | 0.39 | | | 5700 |

We present three different experimental posterior sampling problems across different scientific domains: a simple pendulum, a hydrology simulation that was used by prior methods as a benchmark, and a cosmology simulation that stresses computational limits. Because of the computational expense of the cosmology simulation, we are not able to collect a sufficient number of samples to get a reliable estimate of the effective sample sizes. Instead, we compare our estimates to those in the cosmology literature.

For each experimental domain, we construct three fidelities by adjusting the relevant simulation parameter. In all cases, we measure our abilities to sample from the highest fidelity ($j=0$). For methods labeled "(single)," there is a single higher fidelity layer ($J=1$). For methods labeled "(double)," there are two higher fidelity layers ($J=2$): the one from the "(single)" experiments, plus one more that is even more coarse. For the coarsest fidelity, a Gaussian proposal distribution is used with an adaptive covariance matrix. In the pendulum and cosmology experiments, this normal distribution is reflected to keep parameters within their respective ranges. Shrek MCMC can be extended beyond $J=2$ layers. However, just two layers improves over the standard MCMC and other multi-level methods significantly. Layers' costs should be roughly orders-of-magnitude different in computational costs. For these examples, $J=2$ is the limit of how many layers can practically be constructed with orders-of-magnitude different computational costs.

## 4.1 Simple Pendulum

The equation of motion for a pendulum of length $L$, mass $M$, and initial angle $\alpha_0$ is $\ddot{\alpha} = -(g/L)\sin\alpha$. Our goal is to sample from the posterior of the distribution the two parameters $\theta = (L, \alpha)$ conditioned on the observations of $\alpha$ at three irregularly spaced times during the motion: $\alpha(1) = -0.85$, $\alpha(2.3) = 0.9$, $\alpha(5.0) = 0.95$. Observations of these angles are assumed to be corrupted by Gaussian noise with known standard deviation: $\sigma = 0.1$. Different fidelities correspond to adjusting the error tolerance of an adaptive Runge-Kutta 4(5) ODE integrator ($10^{-3}$ or $10^{-6}$ in our experiments) using Dormand & Prince (1980) with stepsize control and dense output (Hairer et al., 2000). As a separate coarsest layer of approximation, we use the small angle approximation (which does not hold for the observations), $\sin(\alpha) \approx \alpha$, reducing the equation of motion to a simple harmonic motion which can be solved analytically as $\alpha(t) = \alpha_0 \cos(t\sqrt{g/L})$. The difference between the finest fidelity posterior and this small angle approximation is shown in Figure 2.

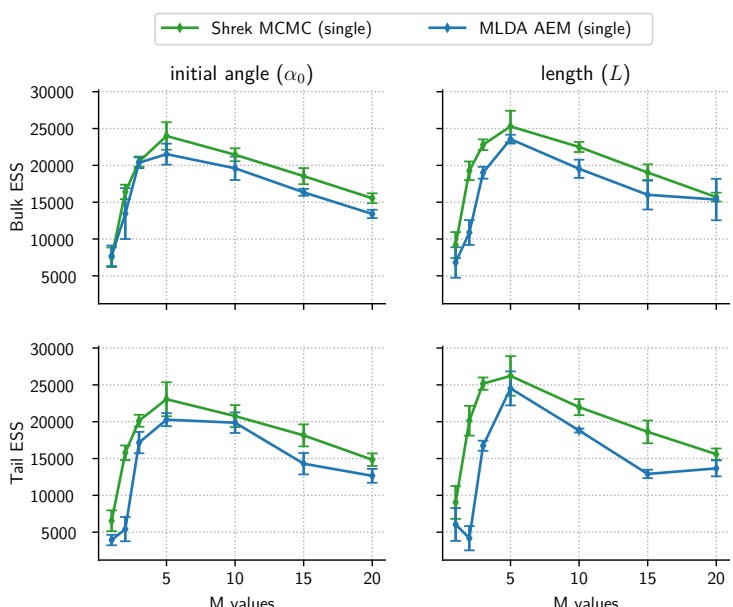

Figure 3: ESS as a function of M for two single layered ($J = 1$) methods with fixed computation time of 500 seconds each ran across 50 different chains.

**Results**  To judge the importance of setting $M$, we evaluated our method across different values of $M$ with fixed total computation time. The effective sample size (ESS) is plotted as a function of $M$ for the two competing multilevel methods in Figure 3. We can see that beyond $M = 5$, the increased computation time from running longer inner subchains leads to a decrease in overall sampling efficiency, indicating that $M = 5$ offers the best trade-off between computation time and sampling efficiency. Therefore, the layered subchains were run for $M = 5$ steps.

We ran 10 chains of the finest fidelity for all methods.We replicated this experiment (of 10 chains) 50 times. Table 1a summarizes the mean effective sample size per second (ESS/s) and the average mean of the parameters across all 500 chains with the standard deviation along with the acceptance rates for every layer.

Figure 4 shows ESS (across all 10 chains) as a function of computation time for each method, with the total number of samples ($N$) generated in 1000 seconds. The standard deviations are plotted as (barely visible) vertical bars. For the sake of readability, we have separated our plots to show how each method performs with one level of nesting (single) and two levels of nesting (double).

We note that the MFMC method obtains significantly more samples in the same time budget. This result arises from its randomized fidelity selection, which collects samples at low-fidelity evaluations more frequently than other methods. The samples from this method are not from the true posterior, but rather can be corrected to estimate an expectation (like the mean). Therefore, we do not list the effective samples in the table, as they are not samples from the true posterior. However, for comparison, the average bulk and tail ESS/s for 50 runs of 10 chains each measured for parameters $[\alpha, l]$ are $[24.80, 20.46]$ and $[26.52, 30.61]$ for MFMC.

All methods are able to improve by using more fidelities. Our Shrek MCMC method is consistently and significantly better than the other methods (including the best one, MLDA with AEM as shown in the table) in terms of ESS/s in both the bulk and tail of the distribution. The acceptance rates for the multilevel

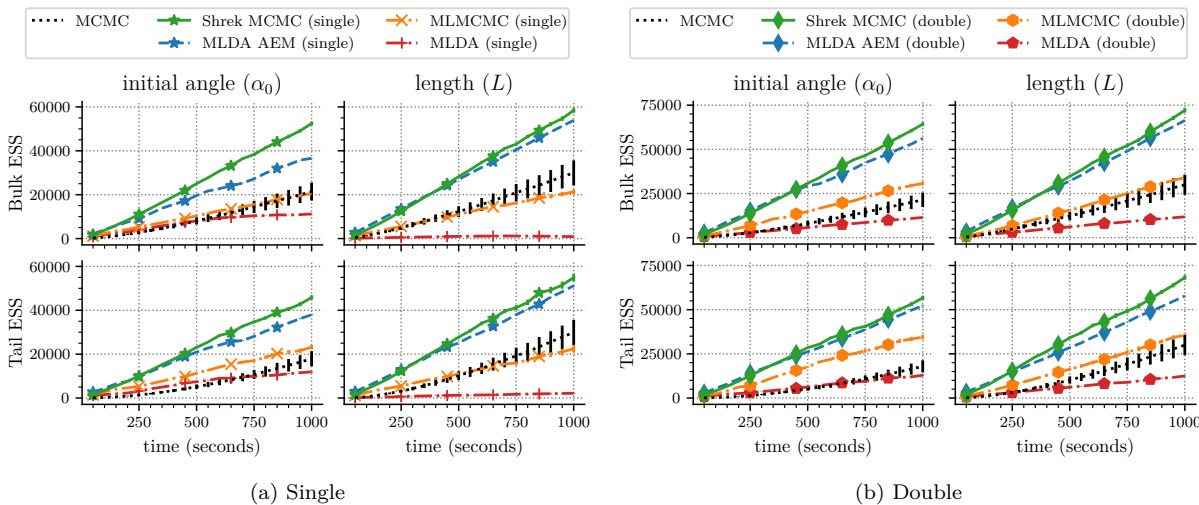

Figure 4: Pendulum Model: ESS for bulk and tail across 50 runs (mean and std. dev.) for single and double layers of nesting.

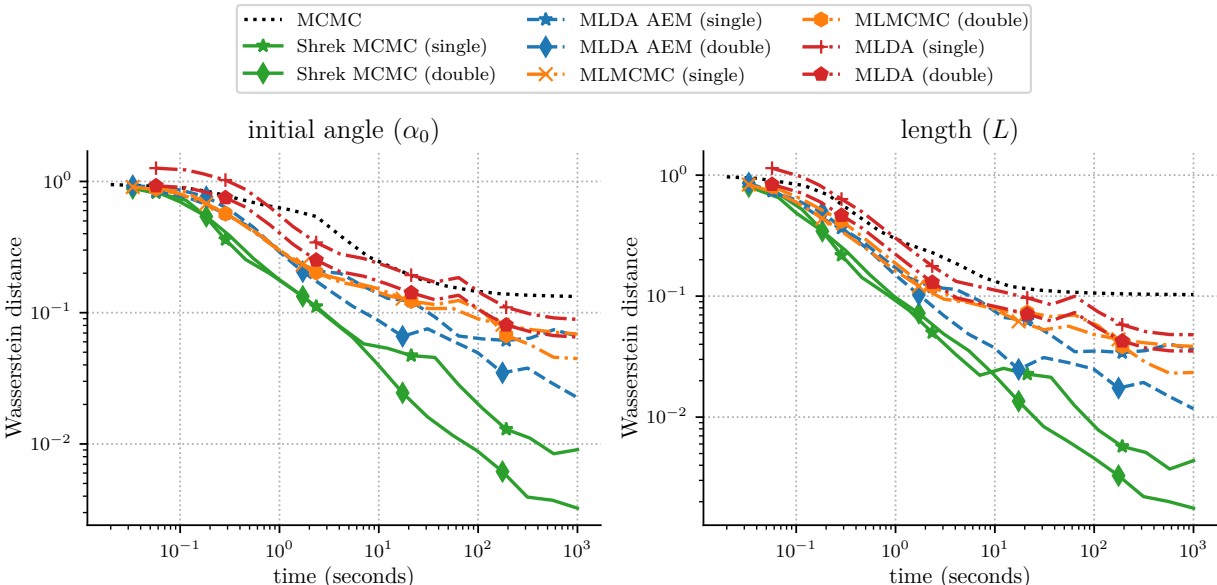

Figure 5: Wasserstein distance to true distributions for the pendulum model.

methods indicate that while the coarsest level accepts about a third of the samples (consistent with Gelman et al. (1996)), the proposed sample from that level is accepted by the finer levels frequently since it was already accepted by an approximate coarse level. The mean of the parameter across different runs of the standard MCMC has a higher standard deviation compared to the other multilevel methods suggesting that some runs of MCMC do a poor job at finding the modes in the posterior. Shrek MCMC produces more samples in the same amount of time compared to its competing method MLDA reflecting that our sampler requires fewer likelihood evaluations per step.

Figure 5 shows the distributional distance to the true distribution as a function of number of samples, averaged over 500 chains. The true mean and standard deviation are unknown for all the experiments in the paper, and measuring the distance between a multi-dimensional distribution which can be evaluated (only

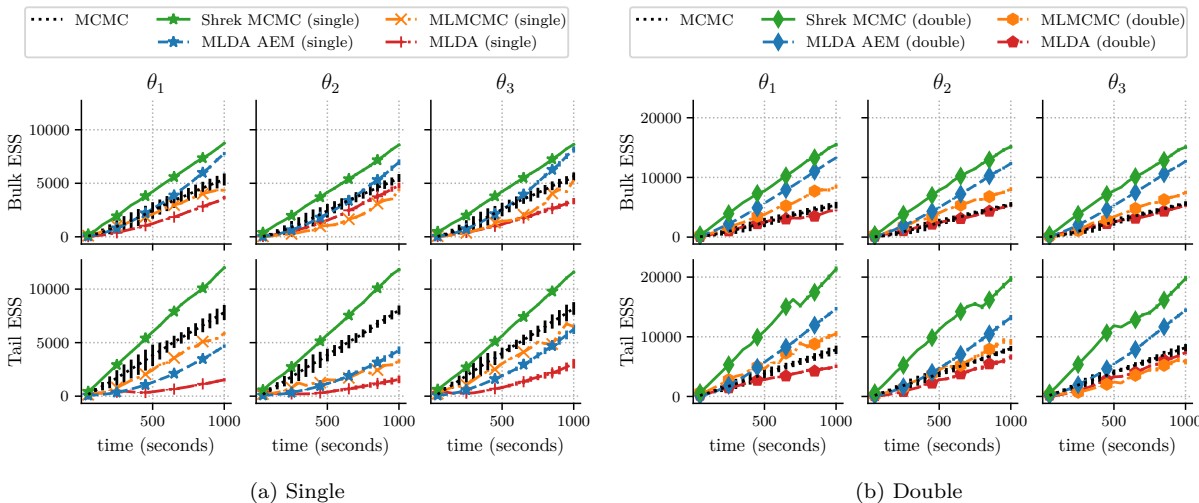

Figure 6: Subsurface Flow Model: ESS for bulk and tail across 50 runs (mean and std. dev.) for single and double layers of nesting.

up to a normalizing constant) and a distribution represented by samples is non-trivial. However, we have analyzed the 1-dimensional marginals of the pendulum model in the following way. We evaluate the true unnormalized distribution on a grid, normalize it, project it to the marginal of interest and then treat it as (weighted) samples for a sample-to-sample Wasserstein distance between it and the samples from the MCMC methods. As we refine the grid, the distances become smaller (for almost all methods). We refine the grid until these distances stabilize, resulting in about a 1000-by-1000 grid (1 million llh evaluations). From the figure, it is clear that for both the parameters, Shrek MCMC ends up with the smallest Wasserstein distance to the true distribution.

## 4.2 Estimation of Soil Permeability in Subsurface Flow

We consider a simple problem in subsurface flow modeling (Dodwell et al., 2015). This model was also used to evaluate the MLDA methods by the original authors (Lykkegaard et al., 2023), and we did not modify the code used by MLDA (except to increase the number of chains and measure time).

The classical equations governing (steady state) single–phase subsurface flow consist of Darcy's law coupled with an incompressibility condition:

$$w + k\nabla p = g \text{ and } \nabla \cdot w = 0 \tag{16}$$

subject to suitable boundary conditions. All quantities are fields over $\mathcal{D} = [0, 1]^2 \subset \mathbb{R}^2$ for these experiments. Here $p$ denotes the hydraulic head of the fluid, $k$ is the permeability tensor, $w$ is filtration velocity (or Darcy flux) and $g$ is the (known) source term.

We are interested in the permeability given observations (with known-variance Gaussian noise) of the hydraulic head at 16 regularly spaced points in $\mathcal{D}$. $k$ is simplified to be the gradient of a random scalar field. The log-Gaussian scalar field is parameterized with a truncated Karhunen-Loéve (KL) expansion (to three terms, following Dodwell et al. (2015)). These three parameters ($\theta$) have a standard normal prior and we sample from their posterior.

Computing the likelihood involves solving a partial differential equation (PDE) with known boundary conditions for a given $\theta$ and comparing the results for $p$ at the observation points. The fidelities correspond to different grid resolutions for the PDE solver: $120 \times 120$ (highest), $30 \times 30$, and $10 \times 10$ (coarsest).

**Results**  Previous work reports that $M = 5$ achieves the best trade-off between effective sample size and computation time for this experimental setup (Lykkegaard et al., 2023). Therefore, we adopt the same value to ensure a fair comparison with our method. Table 1b summarizes the same statistics for this model with the same set-up as the pendulum experiments. Figure 6 shows ESS (across all 10 chains) as a function of computation time for each method. In term of ESS/s, our method improves over the standard MCMC and outperforms the multilevel methods for the same amount of likelihood computational budget, especially in the tail of the distribution. All methods converge to similar means of the parameters with low standard deviation among chains. We note that MFMC collects fewer number of samples compared to other methods since each sample requires multiple log likelihood calculations in the same high fidelity to update $K$, leading to significant add up of computational costs. For MFMC, the estimated mean ESS/s for bulk and tail for parameters $[\theta_1, \theta_2, \theta_3]$ are $[0.128, 0.081, 0.710]$ and $[0.227, 0.161, 1.107]$. But, again, the samples from MFMC were never intended to by interpreted as from the true distribution.

### 4.3  Structure Formation in the Universe with N-body Gravitational Simulation

An important problem in modern-day cosmology is to generate theoretical models of the Universe on very large scales (tens of Mpc across) that can be compared to observations. Bayesian inference allows cosmologists to measure quantities of fundamental physics significance, such as the nature of dark energy and dark matter (Peebles, 1980). The theoretical models needed for next generation telescopes, such as EUCLID (Amendola et al., 2018) and the Roman Space Telescope (WFIRST) (Spergel et al., 2013), are based on expensive numerical simulations, some of which require many days of computer time for each evaluation. For such a computationally expensive model, we show the efficacy of Shrek MCMC as compared with the standard Metropolis-Hastings algorithm and MLDA.

One of the most frequently used summary statistics is the galaxy power spectrum, $\boldsymbol{P}_{\mathrm{gg}}$ (a bold $\boldsymbol{P}$ is a power spectrum, not a distribution): the two-point clustering of galaxies in Fourier space as a function of the wavenumber scale, $k$. We use a slightly simplified model for the galaxy power spectrum for (relative) ease of computation. We perform a forward simulation which starts from a given set of cosmological parameters and predicts the galaxy power spectrum. It works by following the evolution of the Universe under the influence of gravity, from its beginnings in an almost uniform density state to the diverse collection of galaxies sitting in dark matter potentials observed today.

We sample from the posterior density of four cosmological parameters:

$\theta_1$ The dimensionless Hubble constant, $h$, which characterizes the Universal expansion rate and thus the recession velocity of distant galaxies. A redshift zero galaxy at distance $d$ Mpc recedes at a speed $v = H_0 d$, where $H_0 = h \times 100\,\mathrm{km\,s}^{-1}\mathrm{Mpc}^{-1}$. Measuring $h$ is of importance to understand dark energy.

$\theta_2$ The dimensionless total matter density, $0 < \Omega_0 < 1$. $\Omega_0$ is the energy density of matter as a function of the critical density. $\Omega_0$ is important because it can be used to infer the density of dark matter.

$\theta_3$ The dimensionless scalar perturbation amplitude, $A_s$, of the primordial fluctuations at the wavenumber $k = 0.05\,\mathrm{Mpc}^{-1}$. $A_s$ is of interest because it connects to the uncertain high energy physics of the Early Universe. Larger values of $A_s$ correspond to a clumpier early Universe and so lead to larger $\boldsymbol{P}_{\mathrm{gg}}$.

$\theta_4$ The dimensionless linear bias, $b$, which is used to shift the amplitude of our simulated matter power spectrum to match the amplitude of the galaxy power spectrum. This is to account for the difference between observed galaxies and dark matter (which is used by the forward model):

$$\boldsymbol{P}_{\mathrm{model}}(\theta) = b^2 \cdot \boldsymbol{P}_{\mathrm{dm}}(h, \Omega_0, A_s), \tag{17}$$

where $b$ is the scale-independent linear bias and $\boldsymbol{P}_{\mathrm{dm}}$ is the simulated dark-matter power spectrum directly computed from the output density field of FastPM.

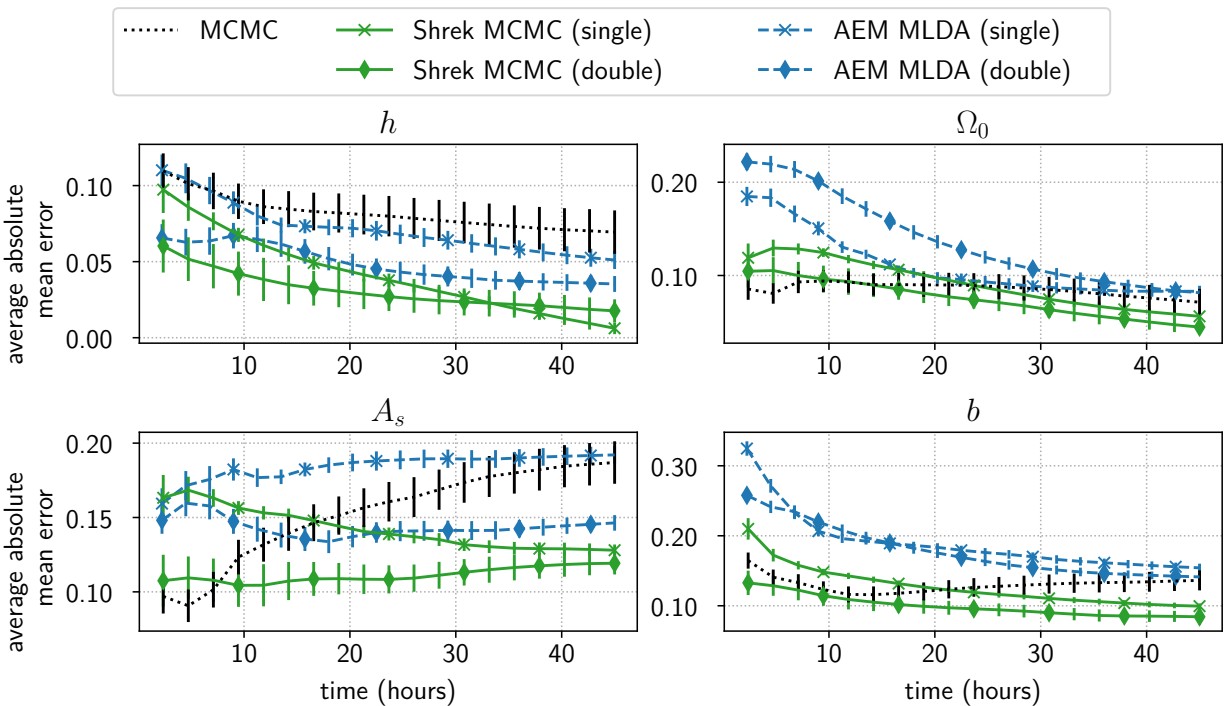

Figure 7: Average (across 10 chains) absolute error in mean estimates for the four parameters, as a function of total simulation time. In the allotted 48 hours, the chains each sampled 800 samples for (plain) MCMC, 600 samples for Shrek MCMC (single), 500 samples for Shrek MCMC (double), 290 samples for MLDA (single) and 200 samples for MLDA (double)

The posterior density is conditioned on the galaxy power spectrum from SDSS-III Baryon Oscillation Spectroscopic Survey (BOSS) Data Release 12 (DR12) as our observational data source (Dawson et al., 2013; Alam et al., 2017). We have used a subset of the BOSS data from the North Galactic Cap (NGC) at $z = 0.38$, which includes $\sim 10^6$ galaxies, from Ivanov et al. (2020).

The likelihood function is a multivariate Gaussian between the galaxy power spectrum from BOSS, $\boldsymbol{P}_{\mathrm{gg}}$, and the galaxy power spectrum from the forward model, $\boldsymbol{P}_{\mathrm{model}}(\theta)$:

$$\ln \mathcal{L}(\theta) = -\frac{1}{2}(\boldsymbol{P}_{\mathrm{model}}(\theta) - \boldsymbol{P}_{\mathrm{gg}})^{\mathsf{T}} \mathbf{C}^{-1}(\boldsymbol{P}_{\mathrm{model}}(\theta) - \boldsymbol{P}_{\mathrm{gg}}) + k. \tag{18}$$

$\mathbf{C}$ is the covariance matrix of the galaxy power spectrum, also estimated observationally.

The most expensive part of the forward model, evolution under gravitational force, is computed using FastPM (Feng et al., 2016). FastPM has a couple of tunable fidelity parameters. The size of the region simulated controls the amount of data available and may have a non-linear effect on the accuracy of the result. We thus fix the size of this region to 1024 Mpc/h and instead change the number of particles. More particles in the simulation mean higher resolution, more accurate power spectrum at higher wavenumber $k$, and thus the likelihood function is higher fidelity. For N-body simulations, the compute time usually scales as $N \log N$, where $N$ is the number of particles. Thus a simulation with a $512^3$ number of particles is $\simeq 80$ times more expensive than a $128^3$ simulation. We therefore set the fidelities by only adjusting the number of particles used in the simulation to be 512 (highest), 384, and 256 (coarsest). Note this calculation is distributed across 20 cores (using MPI) and therefore a saving of 1 hour corresponds to 20 core-hours.

We compare our estimated distributional means to previous computations on the same data. For the parameters $h$ and $\Omega_0$, we compare to the means reported by Ivanov et al. (2020) on the same data using their own MCMC simulation ($h = 0.661$ and $\Omega_m = 0.290$). We have only a single linear bias term, compared with the multiple such terms of Ivanov et al. (2020). Therefore, we can compare neither $b$ nor $A_s$ (which is heavily

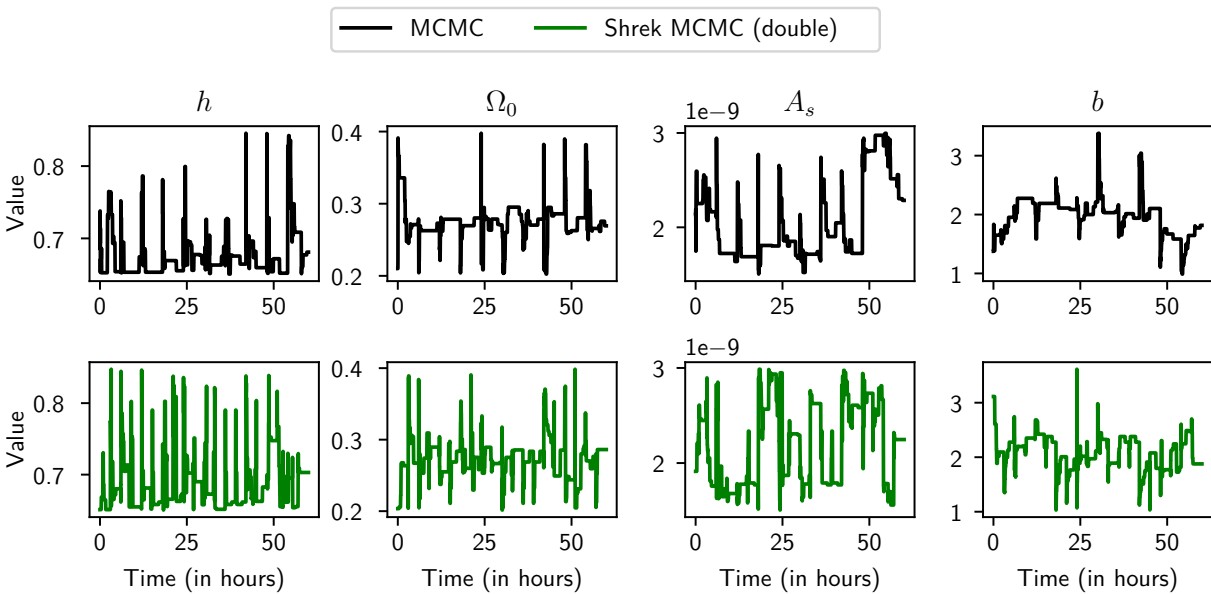

Figure 8: Cosmology model; Trace plot for a random single run of each method

related to $b$) to their results. Instead, we measure $A_s$ against the best fit value from the Planck Satellite (Aghanim et al., 2020), $A_s = 2.09$ and $b = 2$, consistent with comparable BOSS measurements Ivanov et al. (2020). While these are modes (and not means), they are the best independent estimates we can obtain.

**Results** Extreme running time dictated smaller values for $M$ for this experiment. We reduced them by a factor of 2 (approximately) and used $M = 2$ for inner substeps. Figure 7 shows that the Shrek MCMC methods converge to the mean values from previous literature better than the standard Metropolis Hastings method using fewer samples and less time. The $A_s$ parameter has slightly strange behavior. We can still see better convergence of our methods. However, note that the best-fit value of $A_s$ we are taking as "ground truth" is measured (with error) from a different dataset, and thus is likely not the true mean of our posterior. Many large scale structure experiments prefer a lower value of this parameter than Planck, a feature known as the S8 tension (Abdalla et al., 2022). The pairwise plots of the posterior can be found in the Appendix in Figure 10. From the posteriors, it is clear that Shrek MCMC is better at approximating the modes of the distribution as compared to MCMC. Figure 8 shows the trace plot for a random run of MCMC and Shrek MCMC. Our method shows better mixing than MCMC and is less likely to reject proposed samples.

## 5 Summary

Many scientific and engineering problems involve simulations or solving differential equations. In this paper, we present an efficient multi-fidelity layered MCMC that exploits the ability to reduce the accuracy of models leading to approximations of the posterior. In a recursive, nested fashion, these approximations act as proposals for MCMC-based inference. We add layer tuning that successfully encourages the approximate proposals to explore the distribution well. We demonstrate with experimental results using models from three different scientific domains with varying costs that out method, Shrek MCMC, is simple, and yet produces more efficient samples than existing adaptive multilevel MCMC methods with the same computational budget.

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

## A    Convergence Rate Proofs

We show proofs for convergence rates in the main paper. We first use Lemma A.1 to show that after $M$ steps of a coarse chain, we can obtain a minorized lower bound that can be recursively used in its finer layer.

**Lemma A.1.** *Let $p_j(\cdot \to \cdot)$ be the transition distribution of the Markov chain at level $j$ with an invariant target distribution $\pi_j(\cdot)$. For any level $j$, if there exists a $\xi_j > 0$ such that $p_j^1(\theta_j^0 \to \theta_j^1) \geq \xi_j \pi_j(\theta_j^1)$ for all $\theta_j^0, \theta_j^1 \in \Theta$, then*

$$p_j^M(\theta_j^0 \to \theta_j^M) \geq \left(1 - (1 - \xi_j)^M\right) \pi_j(\theta_j^M) \qquad \forall \theta_j^0, \theta_j^M \in \Theta \tag{19}$$

*Proof.* We prove this using induction. We can verify the base case for $k = 1$ such that $p_j^1(\theta_j^0 \to \theta_j^1) \geq (1 - (1 - \xi_j)^1)\pi_j(\theta_j^k) = \xi_j \pi_j(\theta_j^1)$. This is held by the assumption made in the lemma.

Assume using the induction hypothesis that, $p_j^k(\theta_j^0 \to \theta_j^k) \geq (1 - (1 - \xi_j)^k)\pi_j(\theta_j^k)$. We need to show that $p_j^{k+1}(\theta_j^0 \to \theta_j^{k+1}) \geq (1 - (1 - \xi_j)^{k+1})\pi_j(\theta_j^{k+1})$.

Note $p_j^k(\theta_j^0 \to \theta_j^k)$ can be written as $p_j^k(\theta_j^0 \to \theta_j^k) = \phi^k \pi_j(\theta_j^k) + (1 - \phi^k)r_j(\theta_j^k \mid \theta_j^0)$, where $\phi^k$ is the probability that the chain couples to the stationary distribution in $k$ steps, and $r_j(\theta_j^k \mid \theta_j^0)$ is the remaining distribution that depends on $\theta_j^0$.

$$\begin{aligned}
p_j^{k+1}(\theta_j^0 \to \theta_j^{k+1}) &= \int \left[p_j^k(\theta_j^0 \to \theta_j^k) \cdot p_j^1(\theta_j^k \to \theta_j^{k+1})\right] d\theta_j^k \\
&= \int p_j^1(\theta_j^k \to \theta_j^{k+1}) \left[\phi^k \pi_j(\theta_j^k) + (1 - \phi^k)r_j(\theta_j^k \mid \theta_j^0)\right] d\theta_j^k \\
&= \int \phi^k \pi_j(\theta_j^k)p_j^1(\theta_j^k \to \theta_j^{k+1})d\theta_j^k + \int (1 - \phi^k)r_j(\theta_j^k \mid \theta_j^0)p_j^1(\theta_j^k \to \theta_j^{k+1})d\theta_j^k
\end{aligned}$$

We know $\phi^k \geq 1 - (1 - \xi_j)^k$. Thus,

$$\geq (1 - (1 - \xi_j)^k)\pi_j(\theta_j^{k+1}) + (1 - \xi_j)^k \int r_j(\theta_j^k \mid \theta_j^0)p_j^1(\theta_j^k \to \theta_j^{k+1})d\theta_j^k$$

Replacing $p_j^1(\theta_j^k \to \theta_j^{k+1})$ between two consecutive samples with the base assumption,

$$\begin{aligned}
&\geq (1 - (1 - \xi_j)^k)\pi_j(\theta_j^{k+1}) + \xi_j(1 - \xi_j)^k \int r_j(\theta_j^k \mid \theta_j^0)\pi_j(\theta_j^{k+1})d\theta_j^k \\
&= (1 - (1 - \xi_j)^k)\pi_j(\theta_j^{k+1}) + \xi_j(1 - \xi_j)^k \pi_j(\theta_j^{k+1}) \\
&= (1 - (1 - \xi_j)^{k+1})\pi_j(\theta_j^{k+1})
\end{aligned}$$

Therefore using proof by induction, we have that $p_j^M(\theta_j^0 \to \theta_j^M) \geq (1 - (1 - \xi_j)^M)\pi_j(\theta_j^M)$.

$\square$

**Lemma 3.2.** *Assume the minorization condition holds at the innermost level ($j = J$): $p_J^1(\theta_J^i \to \theta_J^{i+1}) \geq \xi_J \cdot \pi_J(\theta_J^{i+1})$ for some $\xi_J > 0$. Then, there exists a minorized lower bound on levels $j < J$ such that*

$$p_j^1(\theta_j^i \to \theta_j^{i+1}) \geq \xi_j \cdot \pi_j(\theta_j^{i+1}) \qquad \forall \theta_j^i, \theta_j^{i+1} \tag{7}$$

*where $\xi_j = (1 - (1 - \xi_{j+1})^M) \cdot \min_\theta \left(\frac{\pi_{j+1}(\theta)}{\pi_j(\theta)}\right)$.*

*Proof.* The transition kernel is given by

$$p_j^1(\theta_j^i \to \theta_j^{i+1}) = \mathcal{A}_j(\theta_j^i \to \theta_j^{i+1}) \cdot q_j(\theta_j^{i+1}|\theta_j^i) + \delta(\theta_j^{i+1} - \theta_j^i) \int \left(1 - \mathcal{A}_j(\theta_j^i \to \theta_j')\right) q_j(\theta_j'|\theta_j^i)\, d\theta_j'$$

$$\geq \mathcal{A}_j(\theta_j^i \to \theta_j^{i+1}) \cdot q_j(\theta_j^{i+1}|\theta_j^i)$$

$$= \mathcal{A}_j(\theta_j^i \to \theta_j^{i+1}) \cdot p_{j+1}^M(\theta_{j+1}^0 \to \theta_{j+1}^M)$$

The sample, $\theta_j^{i+1}$ is proposed using the Mth sample from the $j+1$ chain, therefore is the same as $\theta_{j+1}^M$. Thus, from Lemma A.1,

$$\geq \mathcal{A}_j(\theta_j^i \to \theta_j^{i+1}) \cdot (1 - (1 - \xi_{j+1})^M) \cdot \pi_{j+1}(\theta_j^{i+1})$$

$$= \min\left(1, \frac{\pi_j(\theta_j^{i+1})}{\pi_j(\theta_j^i)} \cdot \frac{\pi_{j+1}(\theta_j^i)}{\pi_{j+1}(\theta_j^{i+1})}\right) \cdot (1 - (1 - \xi_{j+1})^M) \cdot \pi_{j+1}(\theta_j^{i+1})$$

Let $r(\theta) = \frac{\pi_{j+1}(\theta)}{\pi_j(\theta)}$. Then,

$$= \min\left(1, \frac{r(\theta_j^i)}{r(\theta_j^{i+1})}\right) \cdot (1 - (1 - \xi_{j+1})^M) \cdot r(\theta_j^{i+1}) \cdot \pi_j(\theta_j^{i+1})$$

$$= (1 - (1 - \xi_{j+1})^M) \cdot \min\left(r(\theta_j^{i+1}), r(\theta_j^i)\right) \cdot \pi_j(\theta_j^{i+1})$$

$$\geq (1 - (1 - \xi_{j+1})^M) \cdot \min_\theta \left(r(\theta)\right) \cdot \pi_j(\theta_j^{i+1})$$

$$= \xi_j \cdot \pi_j(\theta_j^{i+1})$$

where $\xi_j = (1 - (1 - \xi_{j+1})^M) \cdot \min_\theta \left(\frac{\pi_{j+1}(\theta)}{\pi_j(\theta)}\right)$.

$\square$

**Lemma 3.3.** *For layers $0 \leq j < J$, let $\gamma_j \in \Gamma_j$ be the adaptations for the proposal at layer $j$ or the chain at level $j+1$, i.e, $\gamma_j = \omega_{j+1} \leftrightarrow \psi_{j+1}(\theta) \leftrightarrow p_{j+1}(\theta \to \cdot)$ where $\Gamma_j \in \mathbb{R}$ and $\psi_{j+1}$ is the target at layer $j+1$ with the layer tuning adaptation added. Let $p_{j,\gamma_j}(\theta \to \cdot)$ denote the transition distribution of chain at level $j$ using adaptation $\gamma_j$, starting in state $\theta$. Assume $\forall j, \omega_j \in [\underline{\omega}, \overline{\omega}]$ for some $0 < \underline{\omega} < \overline{\omega}$, and, at the inner most layer, there exists a minorization constant $\xi_J > 0$ such that $\left\| p_{J,\gamma_J}^M(\theta \to \cdot) - \psi_J(\cdot) \right\| \leq (1 - \xi_J)^M$. Then,*

(a) *Simultaneous uniform ergodicity: For all $\tau > 0$. there exists $n = n(\tau) \in \mathbb{N}$ such that*

$$\left\| p_{J,\gamma_J}^n(\theta \to \cdot) - \psi_j(\cdot) \right\| \leq \tau \tag{14}$$

*for all $\theta \in \Theta$ and $\gamma_j \in \Gamma_j$.*

(b) *Diminishing adaptation: The amount of adaptation diminishes in probability with the number of steps t in the adaptation as*

$$\lim_{t \to \infty} \sup_\theta \left\| p_{j,\gamma_j^t}(\theta \to \cdot) - p_{j,\gamma_j^{t+1}}(\theta \to \cdot) \right\| = 0. \tag{15}$$

*Proof.* (a) Consider layer $J - 1$. Using Theorem 3.2,

$$\left\| p_{J-1,\gamma_{J-1}}^M(\theta \to \cdot) - \psi_{J-1}(\cdot) \right\| \leq (1 - \xi_{J-1})^M \text{ where } \xi_{J-1} = (1 - (1 - \xi_J)^M) \min_\theta \left(\frac{\psi_J(\theta)}{\psi_{J-1}(\theta)}\right)$$

From the definition of layer tuning,

$$= (1 - (1 - \xi_J)^M) \min_\theta \left( \frac{(\tilde{\pi}_J(\theta) + \omega_J) \cdot \zeta_{J-1}(\omega_{J-1})}{(\tilde{\pi}_{J-1}(\theta) + \omega_{J-1}) \cdot \zeta_J(\omega_J)} \right)$$

$$= (1 - (1 - \xi_J)^M) \min_\theta \left( \frac{(\tilde{\pi}_J(\theta) + \omega_J)}{(\tilde{\pi}_{J-1}(\theta) + \omega_{J-1})} \right) \left( \frac{Z + \omega_{J-1} \cdot V}{Z + \omega_J \cdot V} \right)$$

where $\zeta$ is the normalizing constant of the new distribution that depends on $\omega$, $Z$ is the normalizing constant of the original distribution, and $V$ is the volume of $\Theta$. With the bounds for $\omega$ from the assumption,

$$\geq (1 - (1 - \xi_J)^M) \min_\theta \left( \frac{\tilde{\pi}_J(\theta) + \underline{\omega}}{\tilde{\pi}_{J-1}(\theta) + \overline{\omega}} \right) \left( \frac{Z + \underline{\omega} \cdot V}{Z + \overline{\omega} \cdot V} \right)$$

$$\triangleq \overline{\xi}_{J-1}$$

By induction with base case at layer $J$,

$$\forall j, \left\| p_{j,\gamma_j}^n(\theta \to \cdot) - \psi_j(\cdot) \right\| \leq (1 - \xi_j)^n \text{ where } \xi_j \geq \overline{\xi}_j . \tag{20}$$

We need to show that for all $\tau > 0$. there exists $n = n(\tau) \in \mathbb{N}$ such that

$$\left\| p_{j,\gamma_j}^n(\theta \to \cdot) - \psi_j(\cdot) \right\| \leq \tau$$

for all $\theta \in \mathcal{X}_j$ and $\gamma_j \in \Gamma_j$.

From Equation 20, we want

$$(1 - \xi_j)^n \leq \tau$$

$$n \geq \frac{\ln \tau}{\ln(1 - \xi_j)}$$

Since $\ln(1 - \xi_j) \leq \ln(1 - \overline{\xi_j})$,

$$n \geq \frac{\ln \tau}{\ln(1 - \overline{\xi_j})} .$$

Thus, for all $\tau > 0$, there exists $n = \max_\tau \frac{\ln \tau}{\ln(1 - \overline{\xi_j})}$ such that $\left\| p_{J,\gamma_J}^n(\theta \to \cdot) - \psi_j(\cdot) \right\| \leq \tau$.

(b) At every step $t$, the change in $\gamma_j$ maps to change in $\omega_{j+1}$. Diminishing adaptation is guaranteed by a gradient descent algorithm with diminishing stepsize that updates $\omega_{j+1}$ at each layer to minimize the Kullback-Leibler divergence between layers $\psi_j$ and $\psi_{j+1}$. At each step of the GD algorithm, $\omega_{j+1}$ is updated as $\omega_{j+1}^{t+1} = \omega_{j+1}^t - \eta_i \frac{\partial}{\partial \omega_{j+1}} H_{j+1}$. To get diminishing adaptation, the update needs to converge as

$$\lim_{t \to \infty} \left\| \eta_t \frac{\partial}{\partial \omega_{j+1}} H_{j+1} \right\| \approx 0.$$

Since we bound $\gamma_j \leftrightarrow \omega_{j+1}$ away from zero,

$$\frac{\partial}{\partial \omega_{j+1}} H_{j+1} = \frac{1}{\tilde{\pi}_{j+1}(\theta_{j+1}^0) + \omega_{j+1}} - \frac{1}{\tilde{\pi}_{j+1}(\theta_{j+1}^M) + \omega_{j+1}}$$

$$\leq \frac{1}{\tilde{\pi}_{j+1}(\theta_{j+1}^0) + \omega_{j+1}}$$

$$\leq \frac{1}{\tilde{\pi}_{j+1}(\theta_{j+1}^0) + \underline{\omega}}$$

$$\leq \frac{1}{\underline{\omega}}$$

Therefore, the update is

$$\lim_{t\to\infty}\left\|\eta_t\frac{\partial}{\partial\omega_{j+1}}H_{j+1}\right\| \le \lim_{t\to\infty}\left\|\eta_t\frac{1}{\underline{\omega}}\right\|$$

$$\le \frac{1}{\underline{\omega}}\lim_{t\to\infty}\eta_t$$

If the stepsize, $\eta_t$ asymptotes to 0, adapation decreases to 0 as $t \to \infty$. Therefore $\lim_{t\to\infty}\left\|\omega_{j+1}^t - \omega_{j+1}^{t+1}\right\| = 0$, and thus $\lim_{t\to\infty}\sup_\theta\left\|p_{j,\gamma_j^t}(\theta \to \cdot) - p_{j,\gamma_j^{t+1}}(\theta \to \cdot)\right\| = 0.$

$\square$

## B    Uniform Smoothing Parameter

Plotted in Figure 9 is the evolution of our tuning parameter $\omega_j$ as a function of samples collected for one example chain of the doubly nested method for the pendulum model. Each sample at $j = 1$ starts a chain of length $M = 5$ at inner layer $j = 2$. The inner layer $j = 2$ uses the small angle approximation of the pendulum as the fidelity. Since it is a poor approximation of the posterior as shown in Fig 2, we start with a relatively high value of $\omega$. This helps the coarsest layer better explore the high-probability regions. As shown, the tuning parameter $\omega$ converges close to zero after a few samples in both the layers. We use a learning rate of $10^{-3}$ for both the layers.

## C    Computational Infrastructure

Our experiments were performed on a machine with 4 Intel® Xeon® Silver 4214 CPUs running at 2.20GHz for our experiments (a total of 48 cores). The machine has 250GB of memory, but memory was never a restriction during our experiments.

All methods use multiproccessing, that is, each chain is run in parallel using a different core. The time listed is across one run of a single chain; however, the effective sample size is calculated across 10 different chains.

For the pendulum and hydrology models, likelihood calculations were carried out on a single core. For the cosmology model, the likelihood calculations were carried out in parallel across 20 cores. Therefore, for the cosmology experiments, saving a day's worth of computation time on the graphs corresponds to saving 20 days worth of core-hours.

## D    Pairwise Plot for Cosmology Model

Plotted in Figure 10 is the pairwise plot for all four parameters of the cosmology model. MCMC generates more samples in the same period of time, yet these samples have not yet converged to the distribution and are still scattered across the space, compared with the relatively compact Shrek MCMC samples.

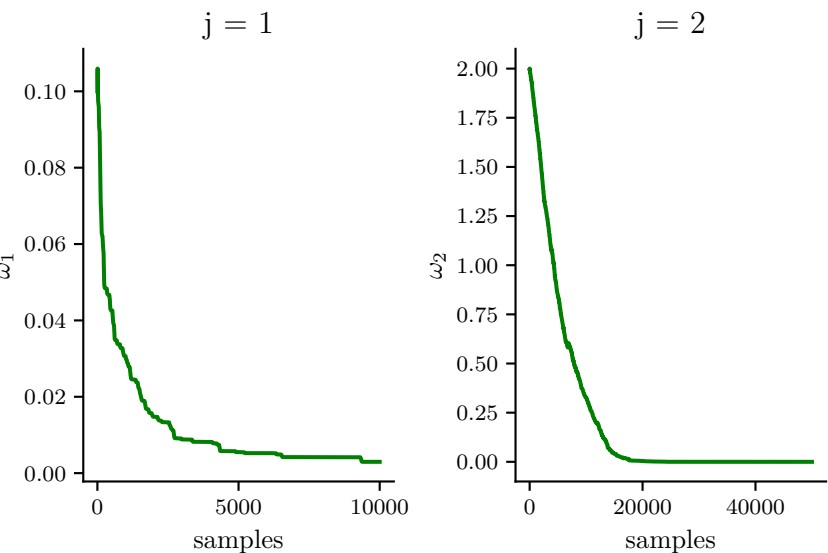

Figure 9: Evolution of $\omega_j$ for doubly nested layers $j = 1$ and $j = 2$

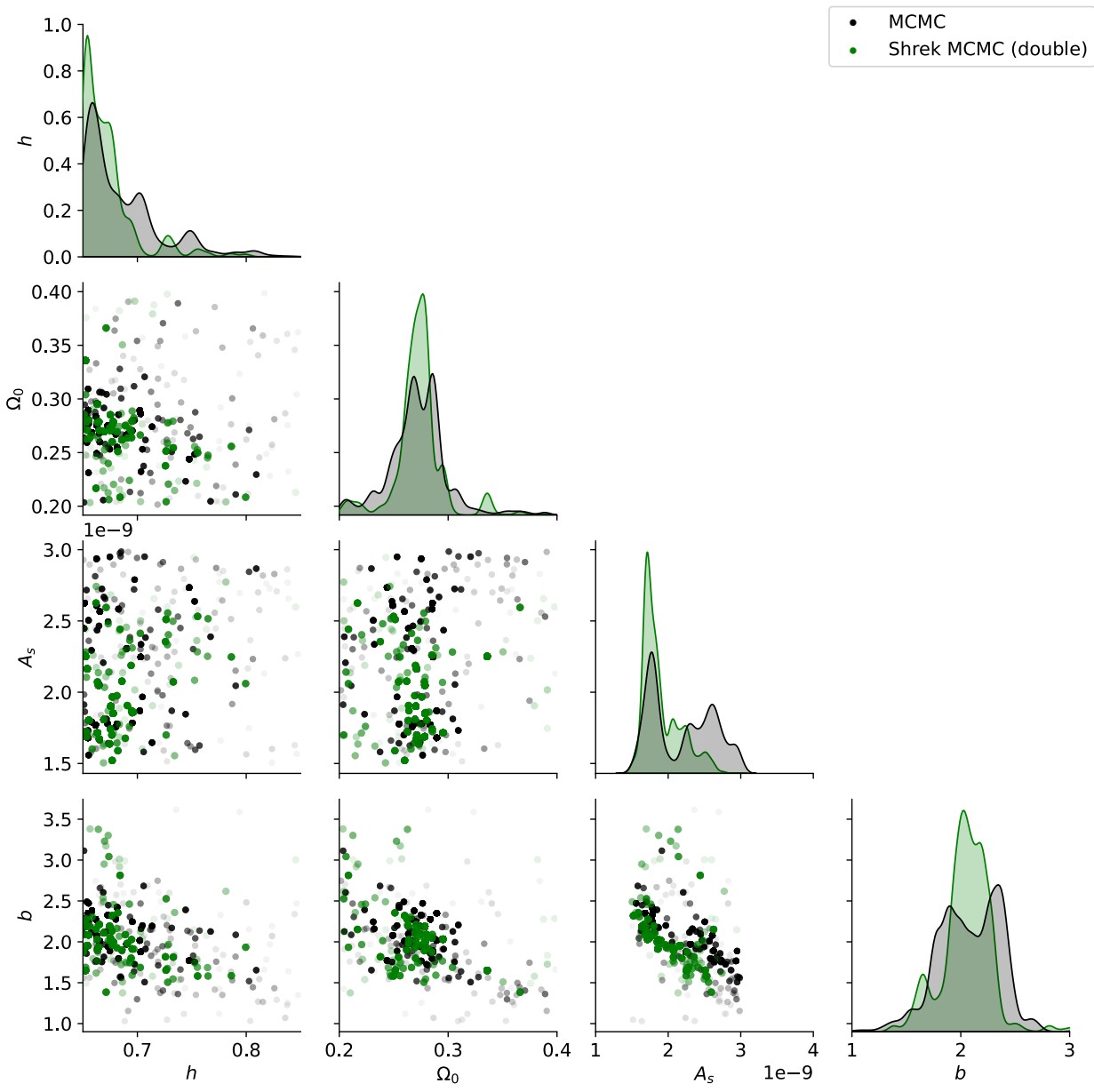

Figure 10: Cosmology model: Pairwise plots for all four parameters. Plotted for a random run of samples collected for 48 hours of MCMC (black) and Shrek MCMC (green).

