# OpenReview forum: "Shrek MCMC: A Multi-Fidelity Layered MCMC Approach"
_TMLR — Rejected by TMLR_

### Review · Reviewer_APFK · 2025-04-01

**Summary Of Contributions:**

## Content

This is a double-blind peer-review and commentary of *Shrek MCMC: A Multi-Fidelity Layered MCMC Approach*;  manuscript TMLR-4345.

The author(s) propose a small modification of a particular instance the MLDA (or (A)TLDA) algorithm in [1], in which state space is the same across all levels, and the inter-level adaptive correction is made by mixing the approximate target at a coarse level with a Uniform distribution on state space to produce the proposal to the next finer level, with the mixture proportion being adapted to minimize KL divergence between distributions at adjacent layers. The lowest-level proposal distribution is the Gaussian set by the standard AM algorithm. A proof of ergodicity is achieved by appealing to the diminishing adaptation condition, following the route introduced in [7], or maybe in an earlier paper by the authors of [7].

Numerical examples are presented with 2 levels (one level of approximation): a pendulum example over $d=2$ dimensional space of parameters; a groundwater problem over a $d=3$ dimensional space; a cosmology example over  $d=4$ dimensional space. As can be seen, these are very low dimensional problems, all on bounded state spaces, though Examples 1 and 3 are notably multi-modal. None of the examples utilize hierarchical modelling for parameters.

### References

- [1] Lykkegaard, M.B., Dodwell, T.J., Fox, C., Mingas, G. and Scheichl, R., 2023. Multilevel delayed acceptance MCMC. SIAM/ASA journal on uncertainty quantification, 11(1), pp.1-30.
- [2] Liu, J.S., 2001. Monte Carlo strategies in scientific computing. New York: Springer.
- [3] Dolgov, S., Anaya-Izquierdo, K., Fox, C. and Scheichl, R., 2020. Approximation and sampling of multivariate probability distributions in the tensor train decomposition. Statistics and Computing, 30, pp.603-625.
- [4] Fox, C., Dolgov, S., Morrison, M.E. and Molteno, T.C., 2021. Grid methods for Bayes-optimal continuous-discrete filtering and utilizing a functional tensor train representation. Inverse Problems in Science and Engineering, 29(8), pp.1199-1217.
- [5] Cui, T. and Dolgov, S., 2022. Deep composition of Tensor-Trains using squared inverse Rosenblatt transports. Foundations of Computational Mathematics, 22(6), pp.1863-1922.
- [6] Christen, J.A. and Fox, C., 2010. A general purpose sampling algorithm for continuous distributions (the t-walk).
- [7] Cui, T., Fox, C. and O'Sullivan, M.J., 2019. A posteriori stochastic correction of reduced models in delayed‐acceptance MCMC, with application to multiphase subsurface inverse problems. International Journal for Numerical Methods in Engineering, 118(10), pp.578-605.

**Audience:**

No

**Broader Impact Concerns:**

## Recommendation

This manuscript has some appealing features and ideas, but ultimately is let down by a multitude of dubious statements that indicate that the author(s) don't have control of the material. The show stopper is that the MCMC that the author(s) propose is basically hopeless, while there are many potentially more efficient options for performing inference in the very low-dimensional problems presented in the examples; the proposed MCMC is not a good algorithmic choice and can't be considered a useful contribution.

While the proof of ergodicity is nice, and necessary, it does not introduce new techniques, and the particular proof has little value when it applies to a poor algorithm.

I recommend that this manuscript be rejected, but that the author(s) be invited to reconsider their algorithm and submission.

**Claims And Evidence:**

No

**Requested Changes:**

## What can be done?

There are quite a few possibilities for designing proposal distributions, or, more generally, for computing inference when the target distribution is multi-modal.

An obvious candidate is to first determine the location of modes -- for example by running the proposed MCMC for a 'tuning' run (10 hours above, looks more than enough) and then use a mixture distribution to fit the target distribution, such as a mixture of (over dispersed) Gaussians with one Gaussian per mode, that is then used as an independence proposal in an MH-MCMC. Probably better is to just use the Uniform proposal, with the coarse model, to find modes. That is essentially a `buckshot' method, followed by kernel density estimation (KDE) with different bandwidths, where the use of radial basis functions is traditional in statistics, and somewhat equivalent. There is lots of literature on such methods. The KDE constitutes a 'surrogate' in the language of MCMC, for which the delayed-acceptance or surrogate transition method [2] are designed.

Maybe better than Uniform at random sampling with KDE is to use a numerical function approximation of the target density (similar to the meshing of the density in the author(s) pendulum example) that is available for these state-space dimensions, as introduced in [3]. Interestingly, that method is applied in [4] to perform inference in a pendulum example that is  more comprehensive than the author(s) pendulum example, and easily copes with multi-modal distributions. An extension of that function-approximation method is in [5], that crosses over to the popular optimal-transport-map methods (see refs in [5]), both methods explicitly approximate multi-modal distributions.

The author(s) might also just look around the literature to find samplers that advertise application to multi-modal distributions, such as [6]. That paper does not appear to address the issue of computational cost, so, presumably, the author(s) would need to develop the method appropriately.

These are just the first papers that I found, treating multi-modal distributions, while searching out from [1]. I can think of many others, such as the `hit and run' or 'simulated tempering' methods from decades ago, that would probably work fine in these low-dimensional settings.

But the advent of deterministic methods (tensor train or transport maps) seems a much better option for computing inference, in these applications, than MCMC that requires many evaluations of an expensive target distribution without remembering or reusing past evaluations. When function evaluations are expensive you should use them to the fullest extent, not just throw them away after an accept/reject step.

## Major minor corrections

A few of the `minor corrections' are actually major,  and indicate that the author(s) have not performed due diligence before submitting this manuscript.

The references are a shambles and need dramatic improvement. Currently the references are far from acceptable, and it appears the authors have not done anything like careful checking before submitting this manuscript.

For example, the reference Lykkegaard et al. (2020) does not have a correct citation to the arXiv paper, or the NeuIPS version, meanwhile there appears to be a more complete published version at https://doi.org/10.1137/22M1476770 that should probably be cited instead.

There are also some papers cited with $\sim$hundreds of authors listed. Such author lists are ridiculous, and should not appear in a citation list. The author lists must be abbreviated in those cases.

## Minor corrections and errors

-  ``Like Shrek MCMC, methods such as simulated tempering and coupled MCMC (Swendsen & Wang, 1986;
Marinari & Parisi, 1992; Altekar et al., 2004) use multiple chains``. \
This is simply false. As shown in Figure 1, Shrek MCMC runs a single chain, because it is a tweak on  MLDA, that runs a single chain, as emphasised in [1]. Also, the Swendsen & Wang algorithm is quite different again.

- It is generally not helpful to the community when authors invent new non-descriptive names, such as ``Shrek MCMC``.  There are already far too many silly non-descriptive names out there. The proposed MCMC is a tweak on MLDA, and should be identified as such.

- The author(s) reproduce some boiler-plate explanations of MCMC methods, regarding HMC, and for Shrek/MLDA/MLMCMC. For example: ``By recursively employing MCMC chains, we can use the coarser resolution models to guide the higher resolution MCMC chain. The result is a sampler for the target model that converges faster and generates more effective samples per computation time, even considering the extra time necessary to employ the lower-fidelity computations.``\
This is the popular-culture statement usually asserted for these methods, but is often not true unless the approximations, and adaptive corrections, are carefully designed. For example, many early applications of multi-level methods, such as the `preconditioned MCMC' due to Efendiev et al, are actually *slower* than the original MCMC, in terms of variance reduction per compute time, yet made almost exactly the same claims as the present author(s). Quantitative evidence is required to back up these claims, otherwise they just add to noise in the literature.

- ``It takes the two-level AEM from Adaptive Delayed Acceptance Metropolis Hastings (Kaipio \& Somersalo, 2007; Cui et al., 2012; 2019)``\
 I wonder if the author(s) have actually read these papers. The approximation error model (AEM) of Kaipio and Somersalo is not adaptive, and is significantly different from the adaptive error model (AEM) of Cui et al. It is unfortunate that someone chose the same acronym. Indeed, the Kaipio and Somersalo AEM converges to the prior distribution, in the limit of training samples, whereas the Cui et al. AEM converges towards the posterior distribution. The former is off-line while the latter is on-line. It would be useful if the author(s) made correct statements and references.

- I noticed that the author(s) jump between Statistics terminology `` unnormalized, normalizing constant ``
and Physics terminology ``partition function``. At one point the author(s) say: ``evaluating the energy of the process and adjusting the temperature`` which is Physics jargon, but never use that terminology again (in the same context).  I think that's potentially confusing. The author(s) should pick the audience and terminology. Personally, I prefer the Statistics terminology because it is more precise than the Physics terminology, and because recent developments in MCMC have used the Statistics language (with a few exceptions).


- ``This effectively mixes the stationary distribution of the jth layer with a uniform distribution (we have added
a constant to the likelihood and then renormalized),``\
The parenthetic comment is not the same as mixing with a Uniform, unless the prior density is constant (w.r.t. the underlying measure). The author(s) say that the prior is Gaussian, not uniform, in Example 2.

- ``Multi Level MCMC (MLMCMC): This method was proposed by Dodwell et al. (2015) and was then
applied to MLDA.``\
I don't think this statement reflects reality. Actually, as pointed out in [1], MLMCMC and MLDA are basically different algorithms.

**Strengths And Weaknesses:**

## What's going on?

There is quite a bit to like about this manuscript, such as: the author(s) using MCMC to compute inference in the cosmology application; that the author(s) have a go at modifying an MCMC to be more efficient for their application; that the author(s) establish ergodicity; that the author(s) adapt both proposal and approximate distributions which is a relatively recent innovation in the literature, so that's nice to see. There are also, unfortunately, signs that the author(s) don't have a broad knowledge of MCMC algorithms, or computational inference more generally, and have actually produced a quite poor method for computational inference in the target applications.

The main content, and substantive problems, in this manuscript can be explained with a few plots:

The first is Figure 2 from the manuscript: (see Fig 2 of manuscript -- you'll have to find that yourself as OpenReview markdown does not support image inclusion; see https://docs.openreview.net/how-to-guides/submissions-comments-reviews-and-decisions/how-to-add-formatting-to-reviews-or-comments)

Note that the distribution being sampled is multi-modal, in both coarse and fine calculations, with the modes being separated by a region of parameter space with essentially zero probability.
This plot is from the 'pendulum example', and unfortunately not the cosmology example (there is no equivalent plot), but I think the same thing is going on.

Second is the trace for one (of four) state variables in the cosmology example, from Figure 7 in the manuscript. (see the left-most lower, green, trace in Figure 7 of the manuscript)

(Yes, the time axis does say hours.) This is one of the better mixed state variables, calculated using the Shrek-MCMC that the author(s) are promoting in this paper, as being efficient.

Thirdly, is a figure that I found on the internet (citation at top of figure): (see Figure 6 in http://dx.doi.org/10.5772/intechopen.82781
)
There are many hundreds of plots like this out there -- I grabbed the first one I found -- and most students learning MCMC will produce a set of  plots like this, to show what a good proposal and mixing looks like (top trace) and what a bad proposal and poor mixing looks like (lower trace, labelled 'improper samples'). The author's plot (green one, above) looks pretty much like the poor mixing case (lower plot), though is maybe marginally better,  which is to say that the author(s) are using a pretty hopeless proposal, for this target distribution; their MCMC is mixing poorly and unlikely to be generating a decent set of samples from the desired distribution. In particular, the MCMC has a high rejection rate and is typically `stuck'.

Also notice the big occasional jumps in the author(s)'s green trace; this suggests that the target distribution in the cosmology example is multi-modal, just like the pendulum example. The authors have used a Gaussian random-walk proposal generated by the AM algorithm (Haario et al., 2001), which converges to a scaled version of the *global* posterior covariance -- not of individual modes --  mixed with a Uniform independence proposal (see Eq. (8)), that is too wide to efficiently sample individual modes, hence the high rejection rate, and does not efficiently move between modes, though is wide enough so there is the occasional big jump (between modes).

That means that the authors have chosen poor proposal distributions for the problems considered, the resulting MCMC is a dog (is mixing very poorly), and the observation that Shrek-MCMC is about 30\% (my rough estimate) faster than other methods (e.g. MLDA) is saying that their hopeless sampler is 30\% better than some other hopeless sampler. Not really worth writing a paper about.

This manuscript, and the author(s)'s work,  suffers from the  basic problem that the target distributions considered are multi-modal, though low-dimensional, while the algorithms chosen -- particularly the proposal distributions -- are ill-suited to this type of distribution.

---

> ### Author Response · Authors · 2025-04-22
> **Reply**
>
> *The cosmology example*
>
> We would like to clarify the cosmology example to better address the reviewer’s concerns.  Previous work, the physical nature of the problem, and Cosmologists all suggest this posterior is unimodal.  It would be quite surprising if it were otherwise, given the set-up.  We believe the features the reviewer is noticing are indicative of a non-Gaussian posterior (particularly in higher dimensions), not a multi-modal one.
>
> Comparing our trace plots to Figure 6 in the paper cited, they superficially look like the “improper samples” examples.  However, this is ignoring the computational time required per sample.  If we compressed our plots along the time axis, they (and even the “improper samples” example in Figure 6) would look like the “correct” plot in Figure 6.  (Any of these plots can be made to look like another if the time scaling is chosen specifically.)  Please recall that an individual likelihood calculation takes 4 minutes.  The long sampling times are an inherent aspect of this example and a primary reason we test on it: There is a real scientific need to speed up calculations on this example (unlike many previous paper’s tests on Gaussians or other toy models).
>
> Most cosmological inference currently relies heavily on Metropolis Hastings.  For example, the recent key paper (https://arxiv.org/pdf/2503.14738) from DESI (the Dark Energy Spectroscopic Instrument) explicitly states they use MH.  The MCMC sampler in the Cobaya likelihood code (https://cobaya.readthedocs.io/en/latest/sampler_mcmc.html) which many cosmologists use is MH.  Perhaps there is a better method, but it is not known.  We have demonstrated how to accelerate this by 30% (or better, depending on the metric used), which is significant, given this is the method used in cosmology.
>
> *Proposal Distributions*
>
> Our purpose is not to hand-craft proposal distributions for individual examples.  Our goal is to use the multi-fidelity nature of many likelihood calculations to automatically generate better (not ideal) proposal distributions (by using an Markov chain that involves the lower fidelity).  There is always a better proposal distribution for a particular example.  Multi-modal distributions are not the focus of this paper.  The two real-world examples are not multi-modal.
>
> There may be other methods, and we are not proposing to limit others’ research.  However, multi-fidelity MCMC has received attention before as a good way of easily improving the efficiency of posterior sampling.  We have a method that out-performs all previous MF methods and comes with some theoretic convergence *rate* bounds.
>
> *Other Points Raised*
>
> We will update the Kykkegaard et al. paper reference.
>
> We will defer to TMLR’s style on how to cite papers with many authors.  We did not want to marginalize the contributions of others by omitting their names in the references section (clearly we do not list all of them in the main paper’s citations).  But, if TMLR would prefer to abbreviate them, we are happy to do so.
>
> Shrek MCMC is more than a “tweak” to MLDA.  The major advance in MLDA (which their paper clearly calls out as such) is the AEM tuning mechanism, and Shrek MCMC fundamentally changes that, which is a major component of its improved performance.
>
> Cui et al.’s own summary directly states that they adapt the AEM of Kaipio & Somersalo.  We cite both to give reference to the line of work that led to AEM being used in Adaptive Delayed Acceptance Metropolis Hastings.  We can move the Kaipio & Somersalo citation to be directly after the word “AEM” to clarify.
>
> We can certainly use “normalizing constant” instead of “partition function” throughout.
>
> We will correct “we have added a constant to the likelihood and then renormalized" to "we have added a constant to the posterior and then renormalized."  The algorithm and code add to the posterior, not the likelihood.  Thank you for pointing out this error.
>
> We will restate that MLMCMC was not “applied” to MLDA, but was “adapted into” MLDA.

---

### Review · Reviewer_McBB · 2025-04-09

**Summary Of Contributions:**

This paper introduces an approach for multi-fidelity MCMC, focusing on the case of the Metropolis-Hastings (M-H) algorithm. The assumption is that the likelihood is expensive to evaluate, and there are a series of models of lower fidelity that are cheaper to evaluate but less accurate (for instance, likelihoods induced by the solution differential equations on a coarser-grained grid). The idea is to run many Markov chains in a nested fashion to help with proposing a new sample. The proposal distribution at a particular iteration and fidelity uses another chain with a lower-fidelity proposal distribution. The paper provides a convergence analysis for the algorithm. Finally, an empirical comparison is performed on a number of problems, including ODE, PDE, and n-body simulation problems.

**Audience:**

Yes

**Broader Impact Concerns:**

I did not see a broader impact statement. I do not have any concerns.

**Claims And Evidence:**

No

**Requested Changes:**

I think the scope of the paper is appropriate for TMLR. However, I believe the paper needs a major rewrite (and review) before it should be accepted for publication, focused on clarity, discussion of the method and related work, and more contextualization of the theoretical assumptions and results.

**Section 1:** The introduction should be rewritten to (1) motivate the problem, and (2) explain how the proposed method addresses the problem (and in particular why it has not already been solved in the literature):
* The paper should make it clear that the primary scope of the article is focused on M-H and not MCMC more generally.
* In paragraph 2: “MCMC is popular because it requires only the ability to evaluate the model’s likelihood” and “it has no additional knowledge of the problem setting to guide its sampling effectively”
This is not quite true for some popular methods (e.g., HMC and Langevin MCMC), and even in the next paragraph, the need for gradients of the log target is mentioned. I understand that the goal is likely to bring up MCMC as a general approach for black-box inference, but I found that the wording itself could be improved.
* In paragraph 3, HMC is brought up and weaknesses such as computing the gradient of the (log) target
* Also change gradient of the “target” to “log target”
* “Something that could be prohibitive” → prohibitively expensive or costly?
* The description of the cosmological simulation used is very generic. It would strengthen the paper to introduce this earlier and explain why it’s a challenging problem. Citations to scientific papers should be included here. In addition, it would be useful to also list broader areas (with citations) in the literature that multi-fidelity MCMC approaches would be useful for, whether or not they are studied in the experiments of this paper.
* The description of Shrek MCMC comes off as very generic and could describe many of the multi-fidelity methods compared against / discussed later in the paper. The recursive part could be emphasized further, as I think that is probably going to be missed by most readers as something new.
* I couldn't figure out where the name "Shrek" arises from. If there's any connection that I missed, this might be useful to let the readers know.

**Section 2**
* For citations on the M-H algorithm, there is more than just Hastings 1970
* Section 2.2 related work: I felt like this section disrupts the "narrative" of the overall paper. The introduction gives very little context for this section due to the genericness of the description of Shrek and the lack of discussion of any most-related multi-fidelity methods. Yet the related work is discussing the most related methods and how they differ from the Shrek algorithm.
Furthermore, Section 2.3 ("Our Contributions") reads more like a statement that should be included in related work (i.e., the part that describes how Shrek differs from MLDA), though part of it could be moved to the introduction.
* In the related work, I think the issue of bias could be discussed a bit more. E.g., there is bias induced from using a two-stage MH algorithm. But some methods use debiasing methods via telescoping sums that require e.g., an infinite number of chains or samples. How do these aspects compare with Shrek? I think it is also worth discussing the larger literature on multi-level / multi-fidelity MCMC, and not just the most relevant ones here. (See comments above.)

**Section 3**
* Figure 1 could be more clear and self-contained. Even after reading the description of the algorithm, I found the figure to be not very helpful.
* The idea of a “layer” vs “fidelity” could be made more clear.
* I am wondering if there is a better way to present Algorithm 1 and 2 – I am not sure Algorithm 1 adds much to the description. It would be useful to provide words to contextualize the variable names and potentially more comments to explain the pseudocode.
* It may be more clear for the narrative to describe the layer tuning algorithm (assuming this is the method being used in the experiments) with the original algorithm, and to describe theoretical results in a separate section.
* After definition 3.2: “The minorization condition of Markov chains can be used to measure convergence rate” – explanation of the minorization condition (along with citations/references) could be included earlier (e.g., Roberts and Rosenthal?), since readers may not be familiar with this paper/result/technique.

**Section 4 and 5**
* Are the Shrek MCMC methods using the layer tuning algorithm?
* The baseline methods have tuning parameters. How are these being tuned?
* I’d be interested to see a discussion of limitations, open problems, future work; e.g., could also discuss connections with other MCMC methods
* Bibliography should be cleaned up before publication

**Strengths And Weaknesses:**

**Strengths**
* To the best of my knowledge, the paper presents a novel approach for Metropolis-Hastings with multiple fidelities. Though I did not see a detailed discussion of this, the method seems fairly general and potentially applicable to some other MCMC methods.
* The paper includes a proposed method, convergence analysis, and a thorough set of experiments. Many papers on the subject do not cover results in all of these areas.
* The inclusion of the n-body simulation problem, where different fidelity simulations can vary across a large number of hours, was an interesting real-world application of the method.

**Weaknesses**
* The biggest weakness to me is that while the high-level idea of the method is clear, the formal presentation of the method itself in Section 3.1 is not very clear. The section on layer tuning is also not very clearly described. (The "claims" box below is primarily about clarity and not the evidence from the experiments.)
* The paper only discusses a small subset of the literature on multi-fidelity MCMC methods; this older survey [1] covers a lot more of the literature, some of which should be discussed, especially methods on delayed rejection (as opposed to delayed acceptance) and multi-stage MCMC methods. Some other papers potentially worth discussing include [2] and [3].
* The title, abstract, and most of the body of the paper says “MCMC” when the focus of the methods, theory, and, experiments is on the Metropolis-Hastings (M-H) algorithm
* The paper seems to lack a formal description of the multi-fidelity problem
* The issue of bias should be discussed more clearly.
* Theory: some of the proofs read more as sketches, than full proofs (e.g. proof of Theorem 3.1 points to “standard Metropolis-Hastings arguments” without any citations to specific theorems and results).
* The main theoretical results in Section 3.3 have limited discussion of assumptions and context. Many results are presented like: “the result is X” without further discussion.
* General writing: the writing lacks polish and precision in several areas.
* There is no discussion of limitations and future directions
* The experiments focus on M-H algorithms: while I think it makes sense to compare the multi-fidelity MCMC with M-H (and other multi-fidelity variants of M-H) as a primary benchmark, I am curious how the method compares with both single and multi-fidelity variants of other MCMC algorithms, which may vary depending on the problem. For example, are the performance improvements from Shrek M-H better than running e.g., single-fidelity or multi-level HMC?


[1] Survey of multifidelity methods in uncertainty propagation, inference, and optimization
https://arxiv.org/abs/1806.10761.

[2] Higdon et al. A Bayesian approach to characterizing uncertainty in inverse problems using coarse and fine-scale information.

[3] Accelerating Asymptotically Exact MCMC for Computationally Intensive Models via Local Approximations https://arxiv.org/abs/1402.1694.

---

> ### Author Response · Authors · 2025-04-22
> **Reply**
>
> Thank you for your suggestions.  We will address all of your suggestions in a revision.  We include a few comments about the most significant ones, below.
>
> *MCMC versus MH-MCMC:* We did, at times, use MCMC when MH-MCMC would be more appropriate.  We will update the introduction (in particular) to correct this.   Past MF MH-MCMC papers have not explicitly called this out in the title, but we can do so.
>
> *Formal statement of problem:* We will complete the statement in Section 3.1 to clarify that the problem is to generate samples from \pi_0.
>
> *Literature review:*  Thank you for including these other papers.  Many of the MCMC sampling papers from the survey are cited in our paper (note that survey covers many other multi-fidelity formalisms), but we will include a discussion of the others.  We will certainly add citations to the others you suggest.
>
> *Other Intro suggestions:*  These are helpful and we are happy to make such edits.
>
> *Section 2 suggestions:*  There certainly *are* more than just Hastings 1970 as citations for MH!  This is the first citation; we can add others.  We needed to explain prior work before our contributions to make the contributions clear.  Placing related work at the end feels disrespectful to all of the prior work done on this problem.  We will consider a different ordering, though.
>
> *Section 3 suggestions:* In discussing this work with a variety of others, it varies a lot whether Figure 1 is helpful.  For some it is not.  For others it is very helpful.  Yes, Algorithm 1 is largely redundant.  We will remove it and clarify how to make the initial call in the text.  We can certainly add comments to explain the pseudo-code.  We specifically do not describe the layer-tuning until later to make the recursive structure clearer.  We’ve found when presented together, it tends to get muddled.  We can certainly expound on the minorization condition.
>
> *Section 4 & 5 suggestions:* Yes, all Shrek MCMC results use the layer tuning.  The baseline methods only have M (number of samples per inner chain).  Those are set the same for all methods.  We use M=5 for the hydrology model, because this is what MLDA (the previous best paper) did for this (their only) example.  We just used the same for the simple pendulum.  For the cosmology model, we picked M=2, acknowledging the large computational effort.  We will add such information.

---

### Review · Reviewer_LtmL · 2025-04-09

**Summary Of Contributions:**

The paper proposes a new multilevel MCMC scheme (Shrek MCMC) that builds on ideas from MLDA but simplifies the structure and implementation. The method leverages recursively defined approximation levels and aims to improve mixing efficiency. Empirical results suggest that the proposed method achieves competitive or superior performance compared to existing competing approaches.

**Audience:**

Yes

**Claims And Evidence:**

No

**Requested Changes:**

- **Tuning guidance:** This is a major gap. The authors should discuss how the algorithm should be tuned, especially when there’s a large discrepancy between coarse and fine models. For example, in the pendulum case, the coarsest model is cheap (ie. orders of magnitude cheaper): should we run many more steps there or not? The authors should propose either practical guidelines or empirical observations. As it stands, it’s not clear that rapid mixing at coarse levels is even beneficial (eg. I think it's clearly beneficial when the coarser model is accurate, but not clear at all to me otherwise)
- **Baseline tuning transparency:** Clarify how competing methods were tuned. Were the same number of steps per level used? Were adaptive proposals used? Without these details, comparisons are difficult to interpret and potentially misleading.

**Non-critical but recommended:**
- **Figure 3 presentation:** Use the same axis for single/double experiments and add a grid to improve readability.
- **Figure/Table consistency:** Reconcile Table 1 and Figure 4. Table 1 suggests AEM MLDA (double) slightly outperforms Shrek MCMC (single), but this isn’t reflected in the figure.
- **Figure 2: mode exploration:** Clarify whether the chains truly explore both modes of the distribution. If not, this may indicate poor mixing.

**Strengths And Weaknesses:**

**Strengths:**
- The proposed MCMC scheme is interesting and novel, and the experimental results suggest strong mixing performance.
- The method is clearly explained and accessible, especially in contrast to MLDA, which is harder to grasp due to suboptimal notation.
- The recursive multi-level structure is appealing and appears competitive with existing multilevel methods.

**Weaknesses:**
- The paper lacks tuning guidance. There is no discussion of how the algorithm should be tuned across levels, which is a key issue, particularly when coarse and fine models differ significantly. A well-mixing coarse-level chain may not help if its samples are far from the target.
- The tuning of baselines is unclear. Details such as the number of steps per level or the use of adaptive Gaussian proposals at coarse levels should be more explicit, I think. It's not obvious that the comparisons reflect well-tuned versions of all methods (since the different methods may require different tuning of the different components)
- The comparison between Shrek MCMC (single) and AEM MLDA (double) is inconsistent: Table 1 suggests the latter performs slightly better, but this isn't evident in Figure 4.
- In Figure 3, using different axes and omitting grids makes the visual comparison harder than it needs to be.
- Figure 2 raises concerns: can the MCMC chains genuinely visit both modes? This should be confirmed or visualized more clearly.

---

> ### Author Response · Authors · 2025-04-22
> **Reply**
>
> *Regarding tuning guidance and transparency:*
>
> There are two tunable parameters.  The first is the adaptation of the proposals.  At the lowest fidelity level, this is the adaptive Gaussian distribution, for which we use a standard method [Haario et al.] which results in no (further) tunable parameters (and is the same for all comparison methods).  For the higher fidelities, we describe the adaptation (via the omega term).  This is tuned using standard gradient descent, as described.  We give the step-size parameters used, although they are not critical.  These adaptations persist for the duration of sampling (and our bounds prove the chains are convergent in this situation).  It is this adaptation that mitigates the problems with using more steps at a lower level that doesn’t (without adaptation) match.  We will make this clearer in the paper.
>
> The second is the parameter of M (number of steps of the inner chain per step of the outer chain). In our experiments, we fix M across all methods to ensure a fair comparison. While it’s possible that more aggressive tuning (e.g., more steps at coarser levels) could help in some cases, our goal was to measure the performance of the methods under the same computational budget.   The same number of steps were used for all multilevel methods. For MFMC, since the algorithm incorporates randomized fidelities, we used the set up proposed by the authors while making sure that the fidelities for each experiment was in a reasonable range.  For the hydrology example, we used the same M=5 value as the previous MLDA method (which used this as the single example to demonstrate its superiority over previous methods).  We kept this for the pendulum example out of simplicity.  For the cosmology example, we let M=2, acknowledging the large computational effort.
>
> We will happily reiterate and make this clearer in the paper.
>
> Previous work did not address how to tune M, and it is not atypical for algorithms to have a tunable parameter.  We agree that better guidance (or automated methods) would be helpful.  To that end, we hope the convergence rate theorem (which relates the convergence rate, M, and the discrepancies between the fidelities’ posteriors) might be a starting point (although clearly does not resolve the issue).  We believe the paper is a significant advance, never-the-less.
>
> *Additional comments:*
>
> The layer tuning parameter is learned using gradient descent after M steps of the inner chain. Since the objective is to minimize the KL divergence between two consecutive layers, the samples from a coarse chain are guided by its finer chain.
>
> Figure 4: Effective samples generally indicate how (un)correlated the samples are.  However, in Figure 4, we are measuring the Wasserstein distance with samples from the “true distribution.” ESS can look large even if samples have not completely explored the distribution (because the ESS calculation has no knowledge of the true distribution), which is why we provide distance/mean comparison to known true values for two of the three experimental results (pendulum and cosmology).
>
> We will add grid lines to the plots to make them readable.

---

> > ### Comment · Reviewer_LtmL · 2025-05-16
> >
> > Thanks for your explanations. Has the manuscript been updated [I do not seem to be able to see the updated version]?
> >
> > > "The second is the parameter of M (number of steps of the inner chain per step of the outer chain). In our experiments, we fix M across all methods to ensure a fair comparison"
> >
> > I am still not entirely convinced that fixing M actually ensures fairness; different methods may be performing best in different regimes. Your statement may be true, and the only way to know is to do some tuning (which is indeed crucial for most MCMC methods).
> >
> > > Previous work did not address how to tune M, and it is not atypical for algorithms to have a tunable parameter
> >
> > While I agree that it is not uncommon to have tuning parameters, it is also often the case that proper tuning is crucial. I still find that the paper, in its current format, lacks in that regard. Have you managed to establish (even empirically) some guidelines for properly tune the different methods?
> >
> > > on my comment: "Reconcile Table 1 and Figure 4. Table 1 suggests AEM MLDA (double) slightly outperforms Shrek MCMC (single), but this isn’t reflected in the figure"
> >
> > Could you please clarify what is going in there, is it just me mis-understanding the claims?
> >
> > To sum-up: one of the main claims of the paper is that "[our method] generates more effective samples in a shorter amount of time and computational cost". For validating this claim, I think that a more robust comparisons with the competition method (ie. demonstrate that competing methods are appropriately tuned) is needed.

---

> > > ### Author Response · Authors · 2025-05-21
> > >
> > > Thank you for the additional comments!
> > > > Has the manuscript been updated [I do not seem to be able to see the updated version]?
> > >
> > > We were not aware that we were allowed to update the manuscript in the middle of review. We have addressed the previous comments (in green) and updated the manuscript.
> > >
> > > > I am still not entirely convinced that fixing M actually ensures fairness; different methods may be performing best in different regimes. Your statement may be true, and the only way to know is to do some tuning (which is indeed crucial for most MCMC methods). Have you managed to establish (even empirically) some guidelines for properly tune the different methods?
> > >
> > > MLDA reported results for the hydrology problem.  For that problem, we used the values of M from the MLDA
> > >         paper, for both their method (because we assumed they had tuned
> > >         it) and for ours (we performed no additional tuning).
> > > For the pendulum (a toy problem), we performed some informal empirical testing that suggests M = 5
> > >         is reasonable.  We have added this ESS as a function of M for that problem to the paper in Figure 3.
> > > For the cosmology model, the extreme running time dictated smaller
> > >         values for M.  We reduced them by a factor of 2 (approximately). We have noted all of this in the revised version now.
> > >
> > > > Could you please clarify what is going in there, is it just me mis-understanding the claims?
> > >
> > > Table 1 reports ESS (the statistic also used in the MLDA paper).  This
> > >         measure is of the samples themselves, and does not reference the
> > >         true distribution (because, in general, this is not calculatable).
> > >         It, essentially, assumes that the samples represent the true
> > >         distribution and seeks to measure their auto-correlation to judge
> > >         how quickly the sampler is generating "effectively new" samples.
> > >
> > > For the case of the toy pendulum model, we can evaluate the
> > >         (unnormalized) llh on a (very fine) grid and normalize to get a
> > >         good estimate of the true distribution.  We can then measure the
> > >         (marginal) Wasserstein distance between the samples and the true
> > >         distribution.  This is a different measure of the effectiveness
> > >         of the sampler.  It does not (directly) measure how quickly
> > >         the sampler "forgets."  But, it does measure how many samples it
> > >         takes to (as a collection) well approximate the true distribution.
> > >
> > > Unfortunately, this Wasserstein measure is not computationally
> > >         feasible for the other problems.

---

### Author Response · Authors · 2025-05-21
**Revision**

Dear reviewers,

We apologize for not being aware that it was permissible to upload a revised version of the manuscript during the review process. We have now uploaded a revised version addressing some specific changes requested by the reviews (made in green for the ease of your review). Thank you!

---

### Decision · Action_Editor_MT6u · 2025-06-09

**Recommendation:** Reject

**Audience:**

No

**Audience Explanation:**

Multi-level MCMC methods are certainly of interest to the community. It is not obvious to me that this specific development is enough of an improvement over past methods to be of interest, but in any case this was not the major determining factor in the final decision.

**Claims And Evidence:**

No

**Claims Explanation:**

This paper presents a multi-level Markov chain Monte Carlo method for sampling from probability distributions. Among other things, reviewers appreciated the clear description of the method and the variety of domains from which the experiments were designed.

However, the reviewers pointed out quite a few flaws in the evidence (both empirical and theoretical) in the paper, and uniformly recommended rejection. Although the authors did not provide a revised manuscript in time for the reviewers to consider it, I did look over the revision myself and did not find the changes adequately satisfied reviewer concerns.

I will emphasize a comment from at least one of the reviewers that the theory is not precise enough to be considered for a TMLR publication. For example, statements like "Shrek MCMC converges to samples from $\pi_0(\cdot)$" have no meaning; and the proof offered is just a rough sketch, not a proof. Some of the definitions have errors (e.g., Definition 3.3). On a more minor note, there are existing, well-known terms and formulations for at least some of the concepts discussed by the authors (e.g., "full support" in Definition 3.1). The above is not a complete list of the technical issues in the paper. I did not spend time checking the correctness of the proofs due to the presence of basic flaws in the main text.

Although this did not impact my decision, I also agree with at least one of the reviewers that "ShrekMCMC" is a poor choice of name for the proposed algorithm. I don't see anywhere in the paper where the name is justified, and it doesn't seem to have any relation to the algorithm itself. I would recommend the authors consider building/relying on names of similar methods already existing in the cited related literature.